# Cryo-EM structure of the human COP1-DET1 ubiquitin ligase complex

Shan Wang [1,2], Fei Teng [1,2,3], Goran Stjepanovic [3], Feng Rao [4] & Ming-Yuan Su [1,2,5] ✉

Ubiquitin modifications regulate fundamental cellular activities by modulating protein stability and function. The ubiquitin ligase COP1, which is present across species from plants to humans, plays a crucial role in the ubiquitination of developmental transcription factors. While COP1 can function independently, it can also be incorporated into CULLIN4-RING ubiquitin ligase (CRL4) complexes through the DET1 adaptor protein. Despite its biological significance, the structural and functional mechanisms of COP1 and DET1-containing complexes remains poorly understood. Here we present the cryo-electron microscopy structures of human COP1 in complex with DDB1-DDA1-DET1 and Ube2e2, revealing an inactive stacked assembly state. Co-expression with COP1 substrates including c-Jun or ETS2 disrupts this configuration, inducing a conformational rearrangement into a distinct dimeric state that allows substrate access. Structural modelling identifies the spatial organization of COP1 WD40 domains where substrate recruits. DET1 serves as a structural scaffold, bridging COP1 and Ube2e2 to initiate potential ubiquitin addition on substrates, while DDB1 recruits the CULLIN4-RBX1 complex to facilitate Ube2d3-mediated ubiquitin chain elongation. These results reveal the dynamic interplay between the structural states of the CRL4$^{DET1-COP1}$ E3 ligase complex and its substrate specific activation mechanism, offering mechanistic insights into ubiquitination regulation and a basis for future studies on E3 ligase dynamics.

The ubiquitin-proteasome system (UPS) is essential for controlling protein quality and cellular homeostasis by mediating the degradation of short-lived regulatory proteins and soluble misfolded or unfolded proteins within the cell. Protein degradation by the UPS is executed through a complex three-step enzymatic cascade involving ubiquitin-activating enzymes (E1), ubiquitin-conjugating enzymes (E2) and ubiquitin ligases (E3). Ubiquitin ligases catalyze the covalent conjugation of ubiquitin to a wide array of substrates in eukaryotic cells. By tagging the substrates with ubiquitin, E3 ligases provide a platform for protein-protein interactions that can modulate the activity, localization or interaction of the target proteins, thereby triggering different biological signals to regulate a variety of cellular processes including cell cycle, cell division, stress response, DNA repair and apoptosis etc[1–4].

CULLIN-RING ligases (CRLs) are the largest family of E3 ubiquitin ligases in eukaryotes, utilizing CULLIN proteins (CULLIN1, 2, 3, 4 A, 4B, 5, 7, 9) as scaffolds[5]. CRL4s consists of a scaffold subunit CULLIN4 and a catalytic subunit RBX1 (RING box protein-1) or ROC1 (Regulator of CULLINS-1) that recruit E2 enzymes[1,6,7]. The adaptor protein DDB1

[1]Department of Biochemistry, SUSTech Homeostatic Medicine Institute, School of Medicine, Southern University of Science and Technology, Shenzhen, China. [2]Key University Laboratory of Metabolism and Health of Guangdong, Southern University of Science and Technology, Shenzhen, China. [3]Kobilka Institute of Innovative Drug Discovery, School of Medicine, The Chinese University of Hong Kong, Shenzhen, Shenzhen, China. [4]Shenzhen Key Laboratory of Biomolecular Assembling and Regulation, School of Life Sciences, Southern University of Science and Technology, Shenzhen, China. [5]Institute for Biological Electron Microscopy, Southern University of Science and Technology, Shenzhen, China. ✉e-mail: sumy@sustech.edu.cn

(DNA damaging-binding protein 1) binds CULLIN4A/B-RBX1 complex and interacts with multiple WD40-repeat proteins (WDRs) to regulate cell cycle progression and DNA damage response[8]. A specific subset of WDRs, known as DDB1-CUL4-associated factors (DCAFs), act as substrate receptors for the CRL4 complex to recruit specific substrates to the complex, facilitating their ubiquitination and subsequent degradation[9,10]. DDA1 (DDB1 and DET1 associated protein 1), initially identified as a component of the plant DDB1-DET1-DDA1 (DDD) complex, which interacts with COP10 and involved in photomorphogenesis repression, was later found to exist in mammals as well[10–13].

Constitutive photomorphogenesis 1 (COP1, also known as RFWD2) is a ubiquitin ligase that is evolutionarily conserved in both plants and mammals. COP1 was first identified in *Arabidopsis* and characterized as a negative regulator of light-mediated development[14]. COP1-based ubiquitination is tightly controlled by photoreceptors and environmental stimuli, enabling dynamic responses to external cues[15,16]. COP1 contains four distinct domains: an N-terminal Glycine/Serine (G/S) rich domain, a RING domain which interacts with E2 ubiquitin-conjugating enzymes; a coiled-coil domain that facilitates dimerization[17], a C-terminal seven-bladed WD40-repeat domain that is responsible for substrate recognition[16,18–20]. In addition, COP1 contains two nuclear localization signals flanking the RING domain, underscoring the importance of COP1 in regulating nuclear processes, likely through substrates ubiquitination and subsequent degradation.

COP1 interacts with the protein De-etiolated 1 (DET1) to mediate substrate degradation in both plants and mammals[18,20–22]. DET1 functions as a recruitment factor for the CRL4 complex by its interaction with DDB1. Together with DDA1, DET1 and DDB1 form a DDD complex. DET1 is a unique member of the DCAF family, as it can directly bind to E2 ubiquitin-conjugating enzymes[17,18]. However, the specific role of the E2 enzymes recruited by DET1 remains unclear. While there are limited reports of DET1 directly interacting with substrates, its primary function appears to be linking COP1 to the CRL4 complex. This interaction may serve as a bridge between two major ubiquitin ligase complexes, facilitating coordinated substrate ubiquitination and degradation. Recent structural studies using cryo-electron microscopy (cryo-EM) have elucidated the structure of the DDD or DDD-Ube2e2 complex, with one report further showing that the assembly and function of CRL4[DET1-COP1] require a network of specific and flexible interactions-DET1 recruits Ube2e family E2s, DDA1 stabilizes DET1 and enables COP1 recruitment, and these coordinated interactions allow the assembly of the functional ligase complex[23,24].

Human COP1 plays a key role in the ubiquitination of a wide range of transcriptional regulators, including p53, c-Jun, and members of the ETS, and C/EBP families[20,25–29]. Among these, the tumor-suppressor p53 was the first identified direct mammalian substrate of COP1. Upon DNA damage, COP1 is inactivated through a process involving its dissociation from p53, ATM-mediated phosphorylation on COP1 S387, autoubiquitination and proteasome mediated degradation[30]. Overexpression of COP1 in cancer cells has been shown to enhance p53 degradation, suggesting a potential oncogenic role for COP1. However, whether p53 is a bona fide substrate of COP1 in vivo is still a topic of ongoing debate[25–27,31]. Similarly, the stability of c-Jun is regulated by COP1, where COP1 is implicated as a tumor suppressor. c-Jun is a member of the AP-1 transcription factor that regulates cell proliferation and differentiation in mammals. In human cells, the assembly of a multi-subunit ubiquitin ligase complex consisting of CUL4, DDB1, RBX1, COP1 and DET1 promotes c-Jun ubiquitination and degradation[19]. Loss of any component of this E3 ligase complex leads to increased stability of c-Jun, enhancing its transcriptional activity. Additionally, serine/threonine kinase 40 (STK40) functions as a COP1 adaptor and promotes the assembly of c-Jun and COP1 complexes[32]. COP1 constitutively maintains low levels of c-Jun in cells, thereby regulating the transcriptional activity of c-Jun/AP-1[33]. Despite the

important role of this E3 ligase complex, it remains unclear how CRL4[DET1-COP1] assembles and how it recognizes the substrates.

In this work, we use single particle cryo-EM to elucidate the structure of human COP1 in complex with DDB1-DDA1-DET1 (the DDD module) and Ube2e2, revealing stacked like assembly state. Our findings show that DET1 serves as a structural scaffold, connecting COP1 to DDD and Ube2e2 enzymes, while DDB1 recruits the CULLIN4-RBX1 complex. Substrates of COP1 including c-Jun or ETS2 could induce conformational change within the assembly. This, in turn, enables engagement of additional E2 enzymes via RBX1 to promote polyubiquitination. The study highlights the dynamic structural interplay within the CRL4[DET1-COP1] E3 ligase complex and its activation upon substrate binding, offering key insights into substrate ubiquitination and the higher-order assembly mechanisms of E3 ligases.

## Results

### Cryo-EM structure of human DDB1-DDA1-DET1 (DDD) complex

To gain structural insights into the assembly of the DDD complex, we co-transfected DDB1, DDA1 and DET1 into Expi293F cells and purified the complex using a tandem affinity purification protocol (Fig. 1A, B, and Supplementary Fig. 1A). The purified DDD complex was subjected to single particle cryo-EM data collection, resulting in a reconstruction with an overall resolution of 2.70 Å (Supplementary Fig. 1, and Supplementary Table 1). Due to the known flexibility of the BPB domain of DDB1 (Fig. 1B), we further performed particle subtraction on this domain to improve the resolution to 2.65 Å, and the resulted reconstruction enabled the fitting of the individual atomic models for DDB1, DDA1 (PDB: 6DSZ) and DET1 (AlphaFold model: AF-Q7L5Y6-F1), allowing for the modelling of most of the complex[34].

The overall structure of the DDD complex demonstrated that the N-terminal helix-loop-helix of DET1 binds to the cleft between the BPA and BPC domains of DDB1, similar to other reported DCAF structures (Supplementary Figs. 2A, B)[35–37]. DET1 exhibited highly flexibility, with several regions missing in the reconstruction, including α5 (residues 162-186), α7-α11 (residues 257-328) and α12 (residues 349-361). The β14–α16 region (residues 329–456) was poorly resolved at higher contour levels and was modelled based on density observed at lower contour levels (Supplementary Fig. 3A). Unlike the classical seven-bladed β-propeller observed in other DCAFs, DET1 adopts a unique architecture. Its blade V is replaced by four helices (α13-α16) while the blade VI contains one helix α17 and two β-strands (β18-β19) (Supplementary Fig. 2C). Consistent with previous structural studies[10,38–40], DDA1 binds to DDB1 via its N-terminus (residues 2-19, 34-49), which wraps around the DDB1 BPA WD40 domain. The C-terminal region (residues 54-67) of DDA1 forms a helix that interacts with DET1, specifically between its blades IV and V.

### Architecture of the human COP1-DDD-Ube2e2 assemblies

COP1 has been reported to form a core complex with the DDD complex. To investigate their interaction, we co-transfected GST-COP1 with the DDD plasmids in Expi293F cells (Fig. 1A, B). After tandem purification using GST and strep affinity tags, we obtained the protein sample for cryo-EM data collection (Fig. 1C). Multiple datasets were collected, including one dataset acquired with a 10° of stage tilt. Four distinct populations were classified from the data: (i) the DDD complex with an overall resolution of 2.93 Å, is highly consistent with previous reconstruction, although additional portions of the DET1 subunit remained unresolved; (ii) the DDD-E2 complex with 2.92 Å resolution, (iii) the dimeric DDD-E2-COP1 complex resolved at 3.03 Å, and (iv) a stacked DDD-E2-COP1 assembly with an overall resolution of about 6.33 Å (Supplementary Fig. 4, and Supplementary Table 2).

Although we successfully obtained the COP1-bound DDD complex, as judged by SDS-PAGE, the two largest populations were those without COP1 bound, suggesting that COP1 dissociates from the DDD complex. After assigning the DDB1, DDA1, DET1 and COP1 (Alphafold

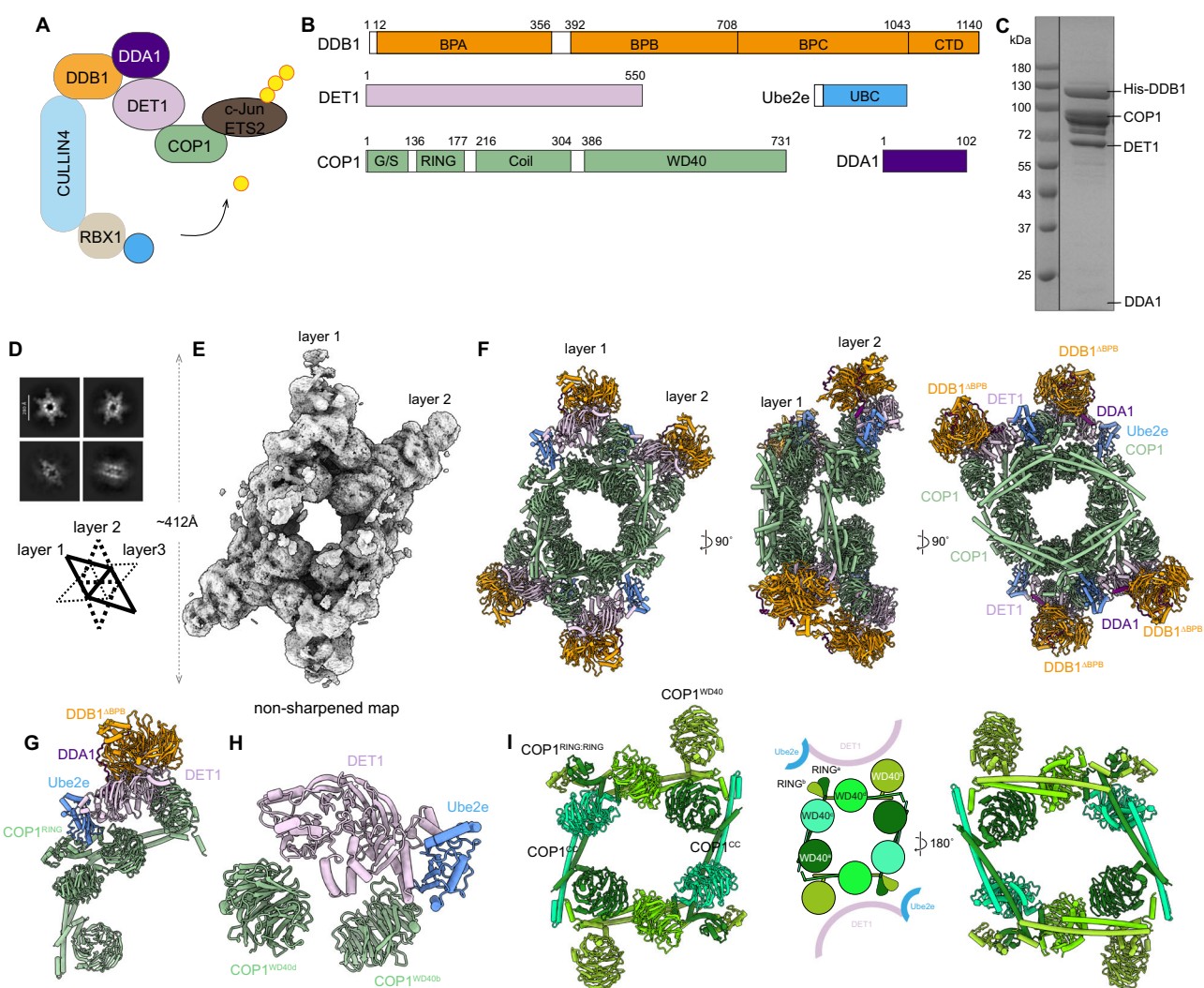

**Fig. 1 | Cryo-EM structure of the stacked human DDB1-DDA1-DET1-COP1 complex. A** Cartoon schematic of the CRL4^DET1-COP1 E3 ligase mediating the ubiquitination of c-Jun or ETS2. **B** Domain architecture of the DDB1-DDA1-DET1-E2-COP1 complex. BPA-BPC, β propeller A-C; CTD, C-terminal helical domain; UBC, ubiquitin-conjugating domain. **C** Coomassie blue-stained SDS-PAGE analysis of the purified DDB1-DDA1-DET1-E2-COP1 complex. The experiment was repeated at least three times independently with similar results. Source data are provided as a Source Data file. **D** Representative 2D class averages of the stacked human DDB1-DDA1-DET1-E2-COP1 complex. **E** The unsharpened cryo-EM map of the stacked DDB1-DDA1-DET1-E2-COP1 complex. **F** Refined coordinates of the stacked complex, visualized in two layers from different views. **G** The refined model of a single DDB1-DDA1-DET1-E2-COP1 complex associated with endogenous Ube2e. **H** Zoomed-in view of the interface between DET1, 2 copies of COP1 and Ube2e2. **I** Close-up of the diamond-shaped arrangement formed by coiled-coil domains among eight copies of COP1 molecules in the stacked DDB1-DDA1-DET1-E2-COP1 complex. CC, coiled-coil.

model: AF-Q8NHY2-F1) molecules in the density map, we observed additional densities whose corresponding protein identities could not be determined (Supplementary Fig. 5A). Mass spectrometry analysis confirmed co-purification of endogenous members of the Ube2e E2 conjugating enzymes family, including Ube2e1, Ube2e2, Ube2e3 and Ube2d3 in the complex (Supplementary Fig. 5B), aligning well with previous report[10]. Notably, the core UBC domains of these proteins were well rigid-body docked into the observed density, guided by secondary structure features (Supplementary Fig. 5A), although the N-terminal extensions of these E2 enzymes remained unresolved[41,42]. Due to the high sequence similarity among Ube2e1, Ube2e2, and Ube2e3, we modeled this region based on the Ube2e2 sequence but removed the side chains (Supplementary Fig. 5C), generating the DDD-E2 (Ube2e2) structure. Interestingly, the previously unresolved region of DET1 spanning α7-α11, which we named the "claw", was found to bind to the backside of the Ube2e family members in the DDD-E2 complex (Supplementary Figs. 5C, D). Furthermore, the α5 and the α12

regions of DET1, as well as β14-α16 (residues 329-456) which were largely unresolved in the DDD-only structure, presumably due to high flexibility, were stabilized via interaction with the claw in the presence of Ube2e protein (Supplementary Fig. 3B, Supplementary Fig. 12).

Intriguingly, we identified some stacked filament-like structures in the sample, which appeared hexagram-shaped in top-view (Fig.1D). Each layer of the filament is formed by twofold symmetric subunits with approximately 412 Å in length, consisting of eight copies of COP1 bound to two copies of the DDD-E2 complex (Fig. 1E–H). Each layer rotates by approximately 50-70° relative to the adjacent top and bottom layers (Fig. 1E, F). To better resolve the structural details, we generated a mask covering the layer 1 or layer 2 and performed masked local refinement (Supplementary Fig. 4). Each layer comprises eight copies of COP1, intertwined through their coiled-coil domains. Specifically, COP1^a and COP1^b dimerize with COP1^c and COP1^d with their coiled-coil domains, forming a RING-RING dimers at the opposite ends of the WD40 domains (Fig. 1I). The WD40 domains of COP1^b and COP1^d

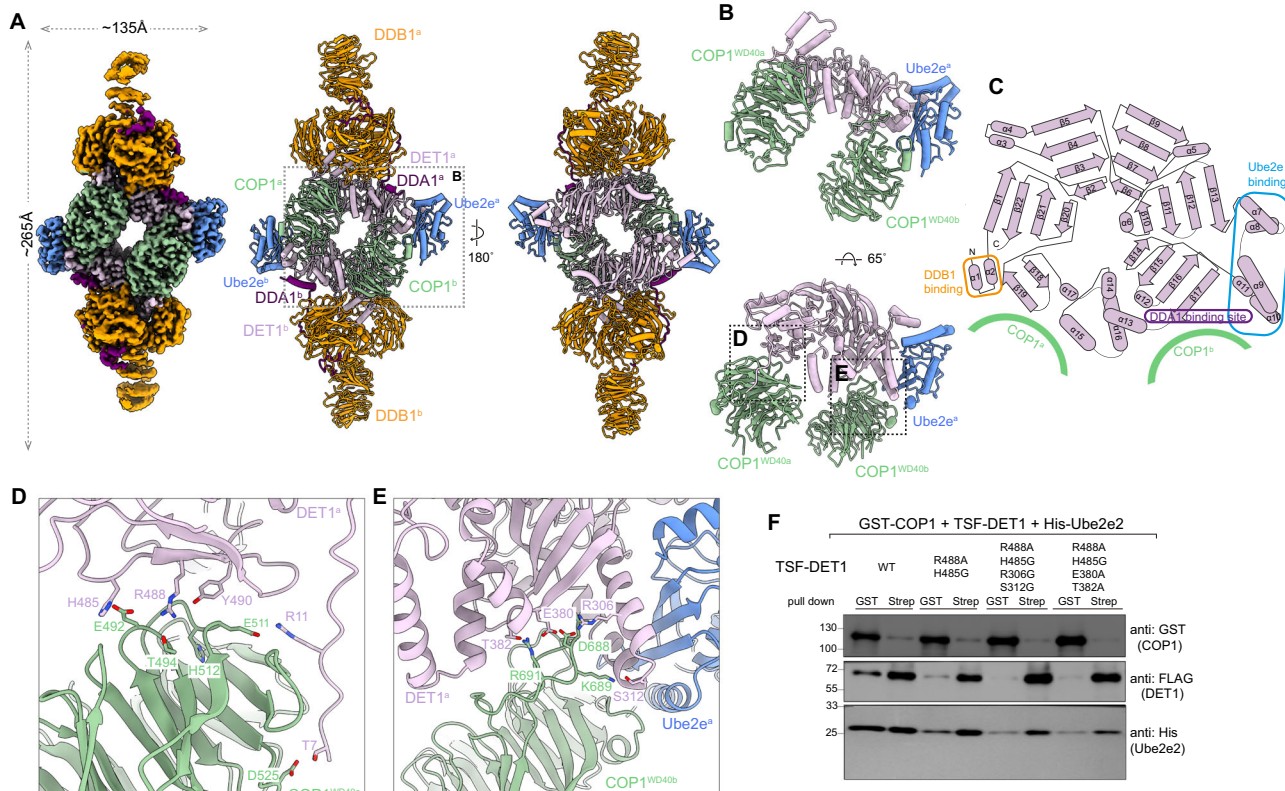

**Fig. 2 | Architecture of the dimeric human DDB1-DDA1-DET1-E2-COP1 complex.** **A** DeepEMhancer post-processed cryo-EM density map (*left*) and atomic model (*right*) of the dimeric DDB1-DDA1-DET1-E2-COP1 complex. The subunits are colored as follows: DDB1, orange; DDA1, purple; DET1, thistle; COP1, dark sea green; Ube2e, cornflower blue. **B** Zoomed-in views of the interface showing DET1 binding to Ube2e and two copies of COP1. **C** Cartoon representation of DET1 highlighting interaction regions with different binding proteins. **D**, **E** Close-up view of the molecular interface between DET1 and COP1$^{WD40a}$ or COP1$^{WD40b}$. The key residues contributing to the interaction are labelled. **F** Pull-down experiment of wild type or key binding residues mutants of TSF-DET1, GST-COP1 and His-Ube2e2, the eluents were analysed by western blotting. TSF, twin-strep-FLAG tag. Data in **F** are representative of three independent experiments. Source data are provided as a Source Data file.

interact with DET1 in a manner similar to the COP1-DDD-Ube2e dimer (described below), while the RING dimers are in close proximity to Ube2e2 (Fig. 1G). Notably, all COP1 WD40 domains in each layer face upward, while their coiled-coil domains are positioned downward, stacking onto the WD40 domains of the COP1 molecules in the subsequent layer. This unique arrangement contributes to the formation of the stacked filament structure.

In addition to the predominant populations of the DDD and DDD-E2 complexes as well as the stacked shaped architecture, we observed symmetric COP1-DDD-E2 dimers with approximately 265 Å in its longest dimension from the cryo-EM analysis (Fig. 2A). The WD40 domain of one COP1 molecule (COP1$^{WD40b}$) binds to blade IV of the DET1 subunit, which is in close proximity to Ube2e2. This suggests that Ube2e2 stabilizes the flexible DET1 claw, which may subsequently associate with COP1. COP1 dimerizes through its coiled-coil domain, allowing another COP1 (COP1$^{WD40a}$), which further interacts with a separate DDD-E2 complex in a similar manner, to associate with blades V-VI of DET1 (Fig. 2B, C). In detail, at the COP1$^{WD40a}$-DET1 interface, the guanidinium group of DET1$^{R488}$ coordinates with the side chains of COP1$^{E492/T494}$. DET1 H485 and R11 form salt bridges with COP1 E492 and E511, respectively. The backbone carbonyl of COP1$^{H512}$ forms a hydrogen bond with DET1$^{Y490}$, while COP1$^{D525}$ engages in hydrogen bonding with DET1$^{T7}$ (Fig. 2D). At the COP1$^{WD40b}$-DET1 interface, DET1$^{S312}$ forms a hydrogen bond with COP1$^{K689}$. The backbone carbonyl of DET1$^{T382}$ interacts COP1$^{R691}$, and DET1$^{E380}$ associate with the backbone of COP1$^{V686/L687}$. Additionally, DET1$^{R306}$ forms a salt bridge with COP1$^{D688}$ (Fig. 2E). Mutations of the interfacial residues in DET1 to glycine or

alanine, including R488A/H485G, R488A/H485G/R306G/S312G or R488A/H485G/E380A/T382A reduced the interaction between DET1 and COP1. This was demonstrated by co-transfection and in vitro reciprocal pull-down assays (Fig. 2F). Together, these interactions result in the formation of a diamond-shaped dimeric DDD-E2-COP1 assembly. However, the N-terminal portions of COP1, including the G/S-rich domains, RING-domain and coiled-coil domain, were unresolved, suggesting that only the WD40 domains are stabilized by interactions with the DDD-E2 complex.

It is well established that COP1 recognizes the substrates through its WD40 domains, specifically utilizing the top surface of its blade IV region[43]. COP1 binds its substrates through their conserved Val-Pro (VP) motifs[43–45]. To explore the substrate-binding sites in both the dimeric and stacked assemblies, we superimposed the COP1$^{WD40}$ with Trib1 (homolog of Drosophila tribbles), which are adaptors for C/EBPα degradation (PDB: 5IGQ) onto the individual COP1$^{WD40}$ domain in these two states of structures (Supplementary Fig. 6A). In the dimeric COP1-DDD-E2 complex, the WD40 domains of COP1 are orientated toward each other and separated by a distance of approximately 20 Å. The substrate-binding pockets are sandwiched between COP1 and DET1, and this spatial arrangement suggests only flexible VP motifs can access the binding pockets (Supplementary Fig. 6B). In the stacked assembly of the COP1-DDD-E2 complex, only the eight WD40 domains in the first layer remain accessible to substrates, while the WD40 domains in subsequent layers are occluded. Upon docking eight Trib1 peptides onto the structure, the substrate-binding sites in the first layer appear more accessible (Supplementary Fig. 6C).

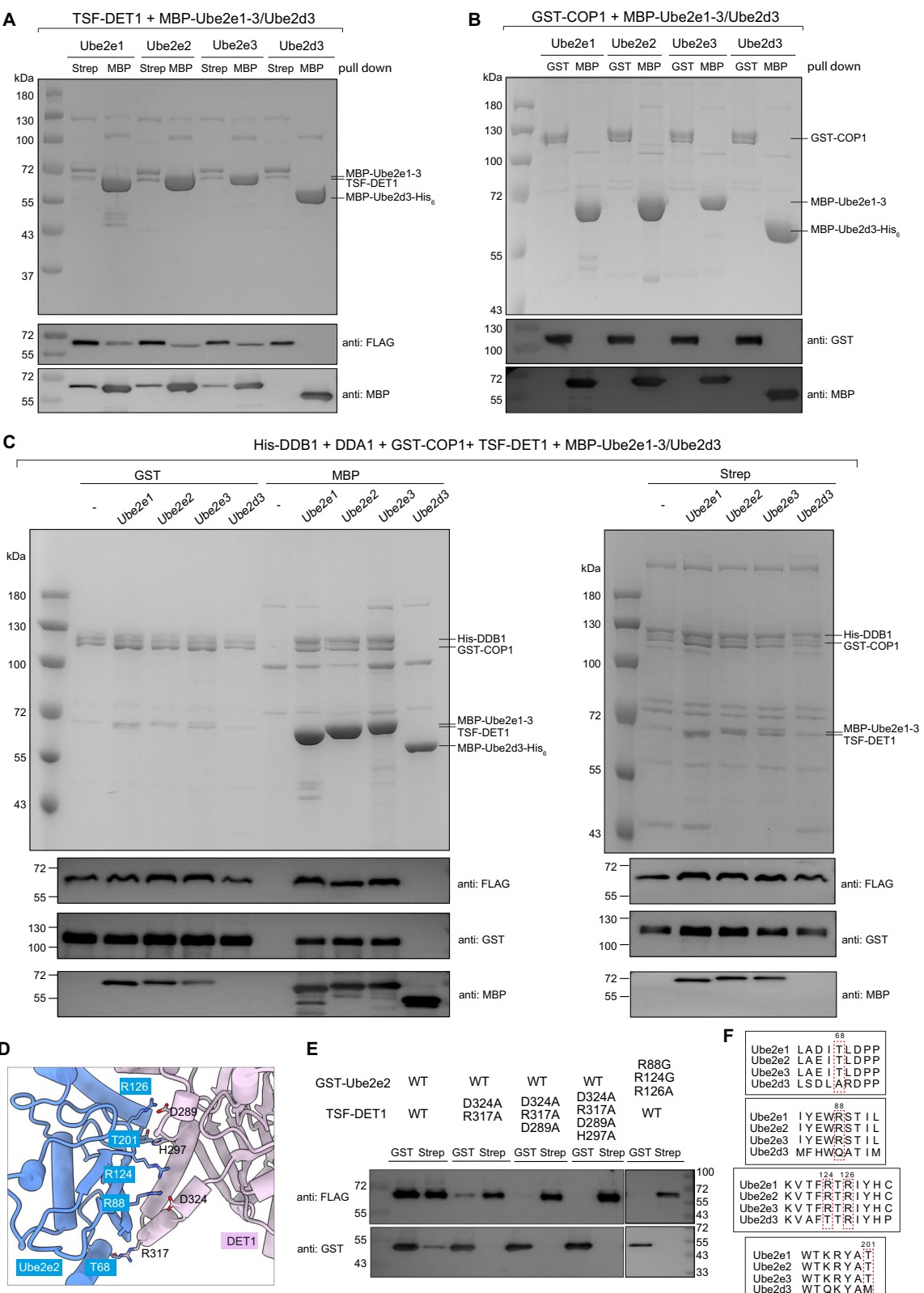

## DET1 serves as a bridge between COP1 and Ube2e family members

Since Ube2e2 is positioned in close proximity to both DET1 and COP1, we investigated whether DET1 or COP1 interact with Ube2e family members by co-transfection and in vitro pull-down experiments. We co-expressed twin-strep-FLAG (TSF) tagged-DET1 and MBP-Ube2e1-3, as well as GST-tagged COP1 and MBP-Ube2e1-3 (Fig. 3A, B). As Ube2d3

was detected in the mass spectrometry, we also examined its potential association with DET1 and COP1. The results demonstrated that TSF-DET1 sufficiently pulled down MBP-Ube2e1-3, but not Ube2d3 (Fig.3A). In contrast, GST-COP1 failed to interact with either MBP-Ube2e1-3 or Ube2d3 (Fig. 3B). Further co-transfection of His-DDB1, DDA1, GST-tagged COP1, TSF-tagged DET1 and either MBP-Ube2e1-3 or Ube2d3 demonstrated that when TSF-DET1 successfully pulled down MBP-

**Fig. 3 | Interaction of the DDB1-DDA1-DET1-COP1 complex with Ube2e1, Ube2e2, Ube2e3 and Ube2d3. A** Pull-down experiment of TSF-DET1 in cells co-transfected with MBP-Ube2e1, Ube2e2, Ube2e3, and Ube2d3. The eluents were analyzed by SDS-PAGE and western blotting. TSF, twin-strep-FLAG tag. Source data are provided as a Source Data file. **B** Pull-down assay of GST-COP1 in cells co-transfected with MBP-Ube2e1, Ube2e2, Ube2e3, and Ube2d3. The eluents were analyzed by SDS-PAGE and western blotting. Source data are provided as a Source Data file. **C** Pull-down experiment of TSF-DET1, GST-COP1, His-DDB1 and DDA1 in cells co-transfected with MBP-Ube2e1, Ube2e2, Ube2e3 and Ube2d3. The eluents were analyzed by SDS-PAGE and western blotting. TSF, twin-strep-FLAG tag. Source data are provided as a Source Data file. **D** Close-up view of the molecular interface between DET1 and Ube2e2, with the key interacting residues labelled. **E** Pull-down experiment of wild-type and key binding site mutants of TSF-DET1 and GST-Ube2e2. The eluents were analyzed by western blotting. TSF, twin-strep-FLAG tag. Source data are provided as a Source Data file. **F** Sequence alignment of Ube2e family and Ube2d3. The residues corresponding to DET1 interaction sites are highlighted with red dashed boxes. Data in (**A**–**C** and **E**) are representative of three independent experiments.

Ube2e2 family members, COP1 was also detected (Fig. 3C). This observation suggests that DET1 interacts with both COP1 and Ube2e proteins, facilitating the presentation of Ube2e proteins by DET1 claw to the COP1 RING dimer. Consistently, the co-transfection assay of COP1-binding defective DET1 mutants including R488A/H485G, R488A/H485G/R306G/S312G and R488A/H485G/E380A/T382A with COP1 and Ube2e2 demonstrated that the reduction of COP1-DET1 interaction resulted in the decreased amount of Ube2e2 co-purification, suggesting that DET1-mediated proximity is not sufficient for COP1 and Ube2e2 to directly interact (Fig. 2F).

Structural similarity analysis using the DALI server suggests that the COP1$^{RING}$-Ube2e2 complex shares a similar organization with the TRIM21$^{RING}$-Ube2e1 complex (PDB: 6FGA) or TRIM21$^{RING}$-Ube2N-Ub (PDB: 6S53) (Supplementary Fig. 7A, B)[46–48]. Guided by sequence alignment between TRIM21 RING and COP1 RING domains (Supplementary Fig. 7C), we generated a COP1$^{RING}$ construct (residues N121-S210). Co-transfection assays of GST-COP1$^{RING}$ and MBP-Ube2e2 demonstrated no direct interaction between them (Supplementary Fig. 7D). These findings further supported that DET1 is essential as a bridge and that COP1 does not engage in a canonical RING-E2 interaction with Ube2e2 (Supplementary Fig. 7D).

We then examined the binding interface between DET1 and Ube2e2. Multiple residues in DET1 form hydrogen bonds and electrostatic interactions with Ube2e2, including DET1$^{D289}$: Ube2e2$^{R126}$, DET1$^{H297}$: Ube2e2$^{T201}$, DET1$^{D324}$: Ube2e2$^{R88}$ and Ube2e2$^{R124}$, DET1$^{R317}$: Ube2e2$^{T68}$ (Fig. 3D). Mutations in DET1 (D324A/R137A) slightly weakened its interaction with Ube2e2, while triple mutant D324A/R317A/D289A or quadplex mutant D324A/R317A/D289A/H297A completely abolished their interaction. Similarly, the corresponding mutations in Ube2e2, R88G/R124G/R124A lost its ability to bind DET1 (Fig. 3E). These interfacial residues in Ube2e2 were identified as highly conserved among Ube2e family members, but were absent in Ube2d3 (Fig. 3F), explaining the specificity and why Ube2d3 did not interact with DET1 in the pull-down experiments.

## Substrate binding of c-Jun induces dimeric DDD-Ube2e2-COP1 formation

Since we hypothesized that the purified stacked DDD-Ube2e2-COP1 protein assemblies might represent inactive states, we next sought to investigate how c-Jun is recognized by the CULLIN4$^{DET1-COP1}$ E3 ligase. Specifically, we aimed to determine whether substrate binding induces conformational changes in the complex. To address this, we co-transfected MBP-tagged c-Jun with the DDD-Ube2e2-COP1 complex and performed tandem purification steps (Fig. 4A). The purified COP1-bound DDD and DDD-Ube2e2-COP1-c-Jun complexes were then analyzed via Superose 6 Increase 10/300 GL gel filtration chromatography (Fig. 4B). For the COP1-bound DDD assembly, two distinct peaks corresponding to stacked filaments and the DDD or DDD-E2 complex were observed (Fig. 4C and Supplementary Fig. 8A). In contrast, analysis of the DDD-Ube2e2-COP1-c-Jun complex revealed three peaks, including 1) aggregates and stacked filaments, 2) dimeric organizations and 3) the DDD or DDD-Ube2e2 complex (Fig. 4D and Supplementary Fig. 8B). These results indicated that co-transfection of c-Jun with the DDD-Ube2e2-COP1 complex in cells leads to the formation of dimeric assembly, as shown by negative-stain EM and gel filtration analyses.

To further explore whether this dimerization process is a substrate-induced effect, we co-expressed another substrate ETS2, with the DDD-Ube2e2-COP1 complex and visualized the resulting assembly (Supplementary Fig. 8C). Similarly, dimeric diamond-shaped particles were observed (Supplementary Fig. 8D), suggesting that substrate binding could trigger conformational changes that transition the complex from a stacked architecture to a dimeric assembly. Additionally, in vitro ubiquitination assays using purified dimeric DDD-Ube2e2-COP1-c-Jun complexes showed c-Jun polyubiquitination in the presence of ATP, E1, Ube2d3 and neddylated CULLIN4-RBX1 (Supplementary Fig. 8E, lane 2), but no when ATP was omitted (Supplementary Fig. 8E, lane 1).

We collected single particle datasets from the purified DDD-Ube2e2-COP1-c-Jun sample (Supplementary Table 3). We identified subcomplexes including the DDD and the DDD-Ube2e2 complex, resolved to 3.25 Å and 3.26 Å resolution respectively (Supplementary Fig.3C), as the two predominant populations, as well as a fraction containing two copies of COP1 bound to one single DDD-Ube2e2 complex, which was resolved to 3.62 Å resolution based on FSC analysis (Supplementary Figs. 9–10). Analysis of the 2D class averages from negative staining, together with the gel filtration elution profile at about 700 kDa, indicating that the DDD-Ube2e2-COP1-c-Jun complex adopts a dimeric architecture. Accordingly, we applied C2 symmetry to generate the structure model (Fig. 4E), and collectively refer to this assembly as the active DDD-Ube2e2-COP1 complex. However, c-Jun itself was not visualized in the cryo-EM reconstruction, suggesting that it is flexible within the assembly.

To further probe c-Jun's association, we performed cross-linking mass spectrometry (XL-MS) using MS-digestable cross-linker disuccinimidyl dibutyric urea (DSBU), which bridges primary amines of lysine residues within ~30 Å (Fig. 4F, Supplementary Table 4). We identified 46 crosslinks within the DDD-Ube2e2-COP1-c-Jun complex, including 22 intermolecular crosslinks, of which 12 are between COP1 and c-Jun. Importantly, these COP1 crosslinks were mainly localized to the WD40 domain, the substrate recruitment region. Together, the XL-MS data support the presence of c-Jun within these protein assemblies, though its flexibility prevents visualization by cryo-EM.

The coiled-coil domain (residues H257-G301) of COP1, which mediates its dimerization, can now be modelled (Fig. 4E). The two COP1 molecules interact with DET1 and Ube2e2 in a similar manner as described before, i.e., their WD40 domains bind to blade IV and blade V-VI of the DET1 subunit, respectively. To further analyze the binding mode of COP1, we examined its interaction with Trib1 (PDB: 5IGQ)[43], where COP1 residues from the top surface of the WD40 domain bind to Trib1 (residues V358-P359) through the bonding between Q356 sits in the binding pocket formed by S425, F645 and W517. Trib1 D355 makes a salt bridge with COP1 K472 and Trib1 Y361 forms a hydrogen bonding with COP1 H578 (Fig. 4G). Mutations of COP1 (residues K472G/H578A/Y491G) abolished its interaction with c-Jun, as demonstrated by co-transfection and reciprocal pull-down assays (Fig. 4H). Based on these results, we concluded the binding mode between COP1 and c-Jun is similar to that of Trib1. Using sequence alignment between c-Jun, Trib1 and other known substrates, we generated a prediction model for c-Jun binding to COP1 (Fig. 4G). Consistent with previous observations, the VP motif (residues 223-241) of c-Jun was predicted to be positioned on

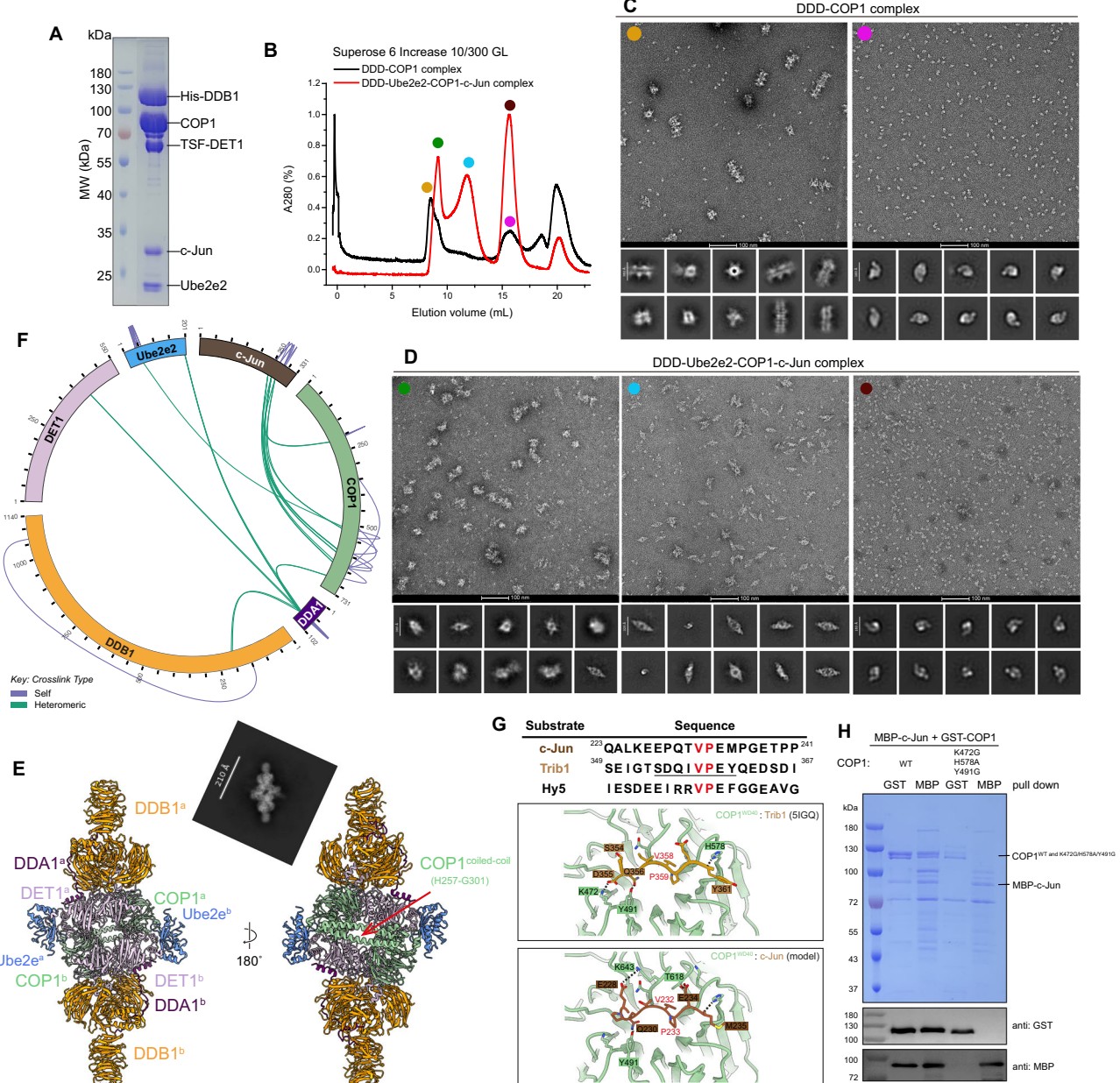

**Fig. 4 | Cryo-EM structure of the human DDB1-DDA1-DET1-COP1-Ube2e2-c-Jun complex. A** Coomassie blue-stained SDS-PAGE analysis of the purified DDB1-DDA1-DET1-COP1-Ube2e2-c-Jun complex. MW, molecular weight. The experiment was repeated at least three times independently with similar results. Source data are provided as a Source Data file. **B** Gel filtration (Superose 6 Increase 10/300 GL) profile of the purified DDD-COP1 (black) and DDD-Ube2e2-COP1-c-Jun complex (red). **C**, **D** The negative stain micrograph and 2D class averages of gel filtration fractions collected from the DDD-COP1 (**C**) and the DDD-Ube2e2-COP1-c-Jun complex (**D**). Scale bar, 100 nm. **E** The chimera model of the human DDD-Ube2e2-COP1-c-Jun complex. The subunits are colored as follows: DDB1, orange; DDA1, purple; DET1, thistle; COP1, dark sea green; Ube2e2, cornflower blue. **F** Circular plot of cross-linked DDD-Ube2e2-COP1-c-Jun subunits identified by XL-MS. Purple lines indicate intra-links, whereas green lines indicate inter-links. Date in F are processed once. **G** Sequence alignment of reported COP1 substrate and adaptor. The conserved VP motif is shown in red. Close-up views of the interface between COP1$^{WD40}$: Trib1(PDB 5IGQ) and COP1$^{WD40}$:c-Jun (model). **H** Pull-down experiment of MBP-c-Jun in cells transfected with wild-type or mutated GST-COP1. The eluents were analyzed by SDS-PAGE and western blotting. Data in **H** are representative of three independent experiments. Source data are provided as a Source Data file.

the top surface of COP1. COP1 Y491 and H578 form hydrogen bonds with c-Jun Q230 and M235 respectively.

It was reported that STK40, a pseudokinase functions as a COP1 adaptor to modulate c-Jun protein levels[49]. Knockout of STK40 could cause the accumulation of c-Jun protein and subsequently impairs mesoderm differentiation[32]. We further investigated whether co-expression of STK40 with the DDD-Ube2e2-COP1-c-Jun complex could stabilize the protein assembly, thereby facilitating substrates modeling. Similar to previous protocol, we carried out tandem affinity

purification and analyzed the protein complexes using negative stain electron microscopy, which exhibits a comparable dimeric architecture (Supplementary Fig. 8E). For single particle analysis, a total of 23,615 micrographs were collected (Supplementary Fig. 11, and Supplementary Table 5). After extensive data processing, the 2D class averages exhibiting dimeric features were subjected to non-uniform refinement, resulting in a reconstruction of 3.7 Å resolution from 274,513 particles. The DDD-Ube2e2-COP1-c-Jun complex co-expressed with STK40 exhibited an overall structure similar to that without

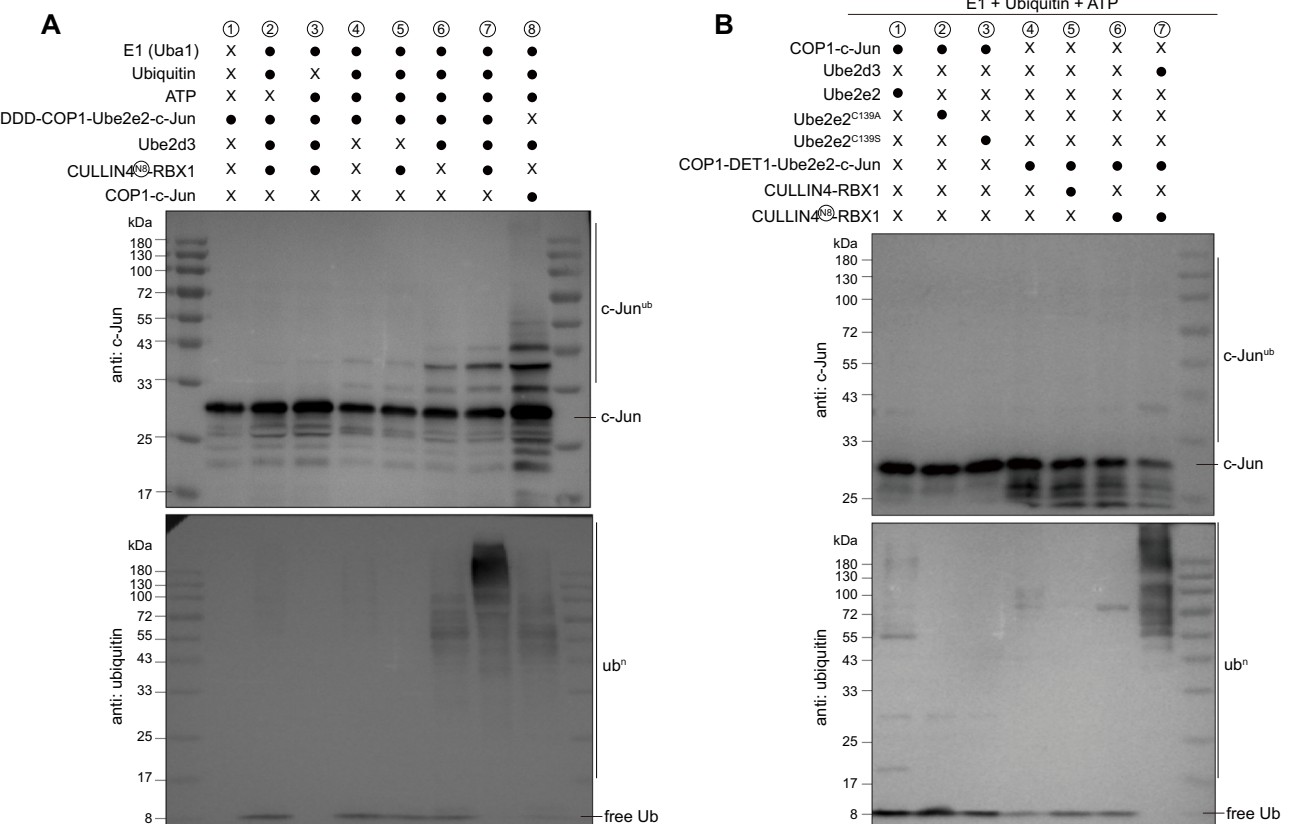

**Fig. 5 | In vitro ubiquitination assay of DDDC-Ube2e2-c-Jun or COP1-c-Jun complexes. A** DDDC-Ube2e2-c-Jun or COP1-c-Jun complexes, purified from Expi293F cells, were incubated with E1 (Uba1), E2, ubiquitin, and neddylated CULLIN4-RBX1. Immunoblot analysis was performed using antibodies against c-Jun and ubiquitin. Source data are provided as a Source Data file. **B** In vitro ubiquitin assay of the COP1-c-Jun complex was performed with E1 (Uba1), ubiquitin, ATP and either Ube2d3, wild type or catalytic inactive Ube2e2 mutants shown on lanes 1-3.

COP1-DET1-Ube2e2-c-Jun complex was incubated with E1 (Uba1), ubiquitin and either CULLIN4-RBX1 or neddylated CULLIN4-RBX1 shown on lanes 4-7. Immunoblotting was performed using antibodies against c-Jun and ubiquitin. The presence (●) or absence (x) of each component in the reaction is indicated. The experiments were repeated independently three times with similar results. Source data are provided as a Source Data file.

STK40, with an additional weak density localized to the COP1 dimerization region (Supplementary Figs. 1I, J). To further characterize this region, we generated a COP1 dimer model using AlphaFold3 prediction and superimposed it to our experimental map[50]. The overall chimera structure displayed a good agreement with the 2D class averages, and the flexible regions of COP1 corresponded to the two protrusions observed in the 2D class averages (Fig. 4E and Supplementary Fig. 1I, red arrows). Nevertheless, neither STK40 nor c-Jun were resolved in the reconstruction, suggesting that these components are highly flexible within the complex. It is possible that STK40 and c-Jun interact with COP1 primarily through their VP motifs, while the remaining regions of these proteins are disordered and therefore not visualized.

## COP1 and DET1 cooperate to promote polyubiquitination of c-Jun

The role of the RING finger in COP1 has been intriguing, as it can independently bind E2 enzymes[51,52], making it appear as a redundant subunit within the CULLIN4[DET1-COP1] complex. COP1 could exhibit ubiquitin ligase activity via its own RING domain towards p53 independently, or in assembly with the CULLIN4-DDB1-RBX1-DET1 complex to degrade c-Jun. However, the precise interplay between COP1 and DET1 is not well understood. In order to investigate the roles of these proteins in the ubiquitination cascade, we performed in-vitro ubiquitination assays using the purified c-Jun bound DDD-Ube2e2-COP1 complex (Fig. 5A). In the presence of E1 (Uba1), ubiquitin and ATP, a higher molecular weight band around 40 kDa of c-Jun was observed,

indicating limited ubiquitin addition (Fig. 5A, lane 4), suggesting that Ube2e2 that interact with DET1 may facilitate the initial modification of c-Jun with COP1 RING domain. The addition of the neddylated CULLIN4-RBX1 complex did not enhance the ubiquitination reaction (Fig. 5A, lane 5), but further supplemented with Ube2d3 which detected from mass spectrometry experiment led to a marked increase in higher-molecular weight c-Jun species, consistent with more extensive ubiquitin addition under these conditions (Fig. 5A, lane 6–7), indicating that Ube2d3 is likely recruited by RBX1 and further elongate the ubiquitin chain on c-Jun. Reactions including COP1-c-Jun with wild-type or catalytically inactive Ube2e2 (Fig. 5B, lane 1–3), as well as COP1-c-Jun with DET1-Ube2e2 complexes (with or without DDB1-DDA1 and either neddylated or non-neddylated CULLIN4-RBX1), did not catalyze c-Jun ubiquitination (Fig. 5B, lane 4–7). Interestingly, the purified COP1-c-Jun complex also yielded detectable c-Jun ubiquitination, although to a lesser extent than reactions containing the full CRL4[DET1-COP1] complex with Ube2d3 (Fig. 5A, lane 8). However, as demonstrated in our co-expression and pull-down experiments, COP1 does not directly interact with Ube2d3. This suggests that Ube2d3 may be recruited by other components via an unclear mechanism to facilitate the polyubiquitination mediated by COP1 RING domains. Taken together, we proposed that DET1 serves as a bridge to recruit Ube2e2, which collaborates with COP1 to initiate ubiquitin on the substrates. Subsequently, DDB1 could recruit the CULLIN4-RBX1 complex to facilitate Ube2d3-mediated ubiquitin transfer and elongation of the ubiquitin chain.

## Discussion

Cullin-RING ligases represent the largest family of multi-subunit ubiquitin E3 ligases and govern various cellular functions. Among these, the CRL4 E3 ligase comprised of the scaffold protein CUL4A/B, the catalytic subunit RBX1 and the adaptor protein DDB1. DDB1 utilizes its flexible BPB domain to bind to the N-terminal region of CULLIN4A/B and interacts with DCAFs which determine substrate specificity. In this study, the DET1 protein functions as the DCAF subunit; however, the substrate recruitment is mediated by another subunit COP1, therefore raising important questions about why COP1 is required to co-act with DET1.

Our findings complemented and extended the recent structural and mechanistic analysis of CRL4[DET1-COP1] assemblies[10,24]. Both studies converge on several key principles. First, DET1 is intrinsically dynamic and adopts conformational states that are stabilized by DDA1 and engagement of the Ube2e2 family. In agreement with Burgess *et al.*, we observed a DET1 claw that embraces the Ube2e2 catalytic core on the backside β-sheet, with the N-terminal Ube2e2 extension remaining flexible. Second, COP1 recognizes substrates via its WD40 domain and engages DET1 through asymmetric WD40-DET1 interfaces, consistent with a COP1 dimer bound to a single DET1. Third, both studies support a division of labor among E2s: Ube2e2 family members are not sufficient to drive substrate ubiquitination with COP1 alone, whereas Ube2d3/UbcH5 enzymes support chain building within CRL4[DET1-COP1] assemblies.

Our work further extended the computational predictions by revealing two higher-order states of COP1-bound DDD-Ube2e2 complexes and their relationship to activity. Specifically, we observed a stacked, filament-like architecture in which only WD40 domains of COP1 in the top layer are accessible, and a substrate-accessible dimeric architecture that is favored upon co-expression of c-Jun or ETS2. The dimeric fraction is catalytically competent in vitro, supporting robust c-Jun polyubiquitination in the presence of Ube2d3 and neddylated CRL4, whereas stacked assemblies are consistent with an autoinhibited state. These observations emphasized the cooperative nature of CRL4[DET1-COP1] and showed enhanced substrate ubiquitination when COP1 is incorporated into CRL4[DET1] assemblies.

COP1 can exhibit distinct oligomerization states, including a dimeric form that is mediated by self-dimerization via its coiled-coil domain. Additionally, COP1 can assemble into an octameric structure by intertwining coiled-coil domain interactions. However, the upstream regulators and the underlying mechanisms that govern these oligomeric states require further investigation. The physiological relevance of the stacked formation also remains unclear. The mechanism by which substrates bind to these regions remains unclear. Could COP1[WD40] bind only a single substrate molecule, or is it capable of simultaneously binding multiple copies? If the central COP1[WD40] bind the substrate, is it further accessible to be tagged with ubiquitin? If a substrate binds to a central COP1[WD40], is the site sufficiently accessible for subsequent ubiquitination? Furthermore, the arrangement of COP1[WD40] domains sealed by layers of organization, suggests a potential unfavourable configuration for substrate ubiquitination. Therefore, we hypothesize that the stacked DDD-Ube2e2-COP1 assemblies exist in an inactive state. It is yet to be determined whether these different assembly states respond to specific cellular signals, facilitate the recognition of distinct substrates, or play a role in assembling with the CULLIN4-RBX1 complex. Whether these structural states are modulated by upstream regulatory signals, such as post-translational modifications or changes in cellular localization, remains to be determined. For instance, glucose regulates CRL4[COP1] assembly[53]. Moreover, phosphorylation events similar to those observed in other E3 ligases could potentially regulate the transition between inactive and active states[54–56]. It was reported that phosphorylation of DET1 Ser458 by MAPK inhibits CRL4[COP1/DET1] by an as yet unknown mechanism[57]. Therefore, we tested if DET1 Ser458 phosphorylation regulates assembly. Using both phosphomimic S458D or phosphor-dead S458A mutants in pull down assays, we observed no disruption of complex formation between DET1, COP1 and the DDD complex (Supplementary Fig. 14). We further repeated the protein co-expression and tandem steps of purifications, i.e. His-DDB1/DDA1/TSF-DET1[S458D]/GST-COP1 to examine the overall conformation of the complex, which is more similar to the wild type complex, and no obvious structural arrangement was detected. Burgess *et al.* reported that the phosphomimetic S458D mutation abrogated COP1 binding in their system, consistent with regulation of DET1 closure. In our hands, S458D did not disrupt DDD-COP1 assembly or architecture by negative-stain EM, although we cannot exclude functional effects in cells or more subtle conformational changes. Differences in constructs, tags, expression systems, or assay formats may underlie this discrepancy. Therefore, further investigate about how Ser458 phosphorylation repressed the activity is required.

Our data also shed light on the role of Ube2d3 in facilitating ubiquitin chain elongation. Although DET1 preferentially interacts with Ube2e family members, Ube2d3 is likely recruited by RBX1 to catalyze polyubiquitination. This stepwise ubiquitination mechanism, involving distinct E2 enzymes for building ubiquitin chain, is reminiscent of other CRL4 complexes but highlights the unique modularity of the CRL4[DET1-COP1] system.

Our findings provide a structural framework to understand how COP1 and DET1 cooperate to degrade key substrates such as c-Jun. This interaction is particularly relevant in the context of cancer biology, where COP1 has been implicated in tumor suppression by regulating transcription factors like c-Jun and oncogenic pathways. The dual role of COP1 as both an independent E3 ligase and a component of the CRL4 complex suggests that it may act as a molecular switch, toggling between distinct functional modes depending on cellular context.

Future research should investigate the interplay of COP1 with a broader range of substrates to determine how structural transitions are fine-tuned for substrate specificity, including p53[26]. COP1-dependent degradation of p53 regulates cancer cell growth and apoptosis. We tested whether COP1 or DDD-COP1 directly interacts with p53 using co-expression and pull-down assays in human cells, and found no detectable interaction under the conditions tested (Supplementary Fig. 15). In contrast, strong binding was observed with c-Jun in parallel experiments. This suggests that direct COP1-p53 recognition may require additional cofactors, adaptors, or specific cellular signal conditions such as glucose induce assembly of CRL4[COP1] for p53 degradation, as suggested in Su et al[29]. Additionally, in vivo studies are necessary to validate the physiological relevance of the structural states identified in this work and their role in regulating cellular processes. High-resolution structures of the entire CULLIN4[DET1-COP1] complex with bound substrates would further illuminate the conformational dynamics that govern ubiquitin chain elongation. Finally, the therapeutic potential of targeting this complex warrants deeper investigation, particularly in cancer models where COP1 activity is dysregulated.

## Methods

### Antibodies

Anti-MBP tag mouse monoclonal antibody and anti c-Jun rabbit monoclonal antibody was used at 1:1000 dilution (Beyotime, AF2912, AF1612), anti-ubiquitin was used at 1:20000 dilution (Abcam, ab134953), anti-GST tag mouse monoclonal antibody was used at 1:5000 dilution (Yeason, 30901ES50), HRP-conjugagted FLAG-tag mouse monoclonal antibody was used at 1:1000 dilution (Yeason, 30502ES60). The secondary antibodies HRP conjugated goat anti-mouse IgG was used at 1:5000 dilution (CWBIO, CW0102S), HRP Conjugated goat anti-rabbit IgG was used at 1:5000 dilution (CWBIO, CW0103S).

## Cloning and mutagenesis

The gene for full-length *COP1* (residue M1-V731) was codon optimized and synthesized by GENEWIZ. The genes encoding *DDB1*, *CULLIN4A*, *RBX1*, *DET1*, *DDA1*, *c-Jun*, *STK40*, *ETS2*, *Ube2e* family (*Ube2e1 - 3*), *Ube2d3*, *Uba3*, *UBC12*, *APPBP1* and *NEDD8* were amplified from cDNA by PCR and subcloned into pCAG vectors with different tags, respectively. For full-length *COP1*, it was constructed as a N-terminal Glutathione S-Transferase (GST) tag followed by a TEV cutting site. *DET1*, *CULLIN4A*, *RBX1* and *STK40* were constructed with N-terminal twin-strep-FLAG (TSF) tag. For *DDB1*, it was cloned with N-terminal His$_6$ tag. For *DDA1*, it was cloned with no-tag or N-terminal GST-TEV tag. For *Ube2e* family members, *c-Jun* and *ETS2*, they were subcloned into a N-terminal MBP-TEV tag. For *Uba3*, *UBC12* and *APPBP1*, they were cloned into pCAG vector respectively. For *NEDD8*, it was cloned with N-terminal HA tag. The *Ube2e2* mutants were generated by two-step PCR and ligated into the pCAG- Maltose-Binding Protein (MBP) or GST vector. The *COP1* mutants were generated by two-step PCR and ligated into the pCAG-GST vector. The *DET1* mutants were generated by two-step PCR and ligated into the pCAG-TSF vector. For in-vitro ubiquitination experiments, *Uba1* was cloned in pGEX-4T1 vector with N-terminal GST tag and C-terminal His$_6$ tag. *Ubiquitin*, *Ube2d3*, *Ube2e2* were cloned into the pGEX-4T1 vector. All constructs were verified by DNA sequencing. The primers used in this study are listed in Supplementary Data 1.

## Protein expression and purification

For the DDB1-DET1-DDA1 (DDD) complex, His-tagged DDB1, TSF-tagged human DET1, and GST-tagged DDA1 were co-expressed in Expi293F cells (Thermo Fisher, A14527CN). Expi293F cells were grown in Union-293 medium and used for protein expression, with 1 mg of total DNA plasmid and 3 mg of PEI (Biohub, 78PEI40000-1g) per 1 liter of cells at a density of $1.5 - 2.0 \times 10^6$ cells/ml. Cell cultures were harvested after 3 days and washed with ice-cold 1× PBS. The cell pellet was resuspended and lysed in lysis buffer (20 mM HEPES-NaOH pH 7.4, 200 mM NaCl, 1% Triton-X100 (v/v), 2 mM MgCl$_2$, 0.5 mM TCEP) with protease inhibitors (1 mM PMSF, 1 μM pepstatin, 4 μM leupeptin, 0.3 μM aprotinin) for 30 min at 4 °C. After centrifugation at $15,000 \times g$ for 50 min, the cell supernatant was incubated with GST beads (Cytiva, Glutathione Sepharose 4B, 17075605) at 4 °C for 1.5–2 h. The beads were washed with wash buffer (20 mM HEPES-NaOH pH 7.4, 200 mM NaCl, 2 mM MgCl$_2$, 0.5 mM TCEP) supplemented with 2 mM ATP. Bound proteins were eluted with wash buffer containing 50 mM reduced glutathione (Sangon, A600229-0050). After TEV digestion overnight at 4 °C, the elution was then subjected to Strep-Tactin Sepharose resin (IBA, 2-1201-010), washed with wash buffer and eluted with wash buffer supplemented with 10 mM desthiobiotin (Sigma-Aldrich, D1411).

For the assembly of DDB1-DET1-DDA1-COP1 complex, His-tagged DDB1, TSF-tagged human DET1, GST-tagged COP1 and untagged DDA1 were co-transfected into Expi293F cells as previously described. After 12 h of incubation, 1 mM sodium butyrate was added, and the cells were further incubated for an additional 48 h. Cells were then harvested by centrifugation and washed with ice-cold 1× PBS. The DDB1-DET1-DDA1-COP1 complex was purified using the same protocol described above.

For the assembly of the DDA1-DET1-DDB1-Ube2e2-COP1-c-Jun complex, untagged DDB1, TSF-tagged human DET1, N-terminal GST- and C-terminal His-tagged COP1, untagged DDA1, MBP-tagged Ube2e2$^{S76G-C139S}$ and MBP-tagged c-Jun were co-transfected into Expi293F cells as previously described. The Ube2e2 S76G and C139S mutations were originally introduced to perform ubiquitin conjugation assays. Following transfection, cells were incubated for 12 h, after which 1 mM sodium butyrate was added, and incubation continued for an additional 48 h. Cells were harvested by centrifugation, washed with ice-cold 1× PBS. The cell pellet was resuspended and lysed in lysis

buffer for 30 min at 4 °C. After centrifugation at $15,000 \times g$ for 50 min, the cell supernatant was incubated with MBP resin (NEB, E8021S) at 4 °C for 1.5–2 h. The beads were washed with wash buffer supplemented with 2 mM ATP. Bound proteins were eluted with wash buffer containing 20 mM maltose. The MBP flow-through was then subjected to GST resin, washed with wash buffer containing 2 mM ATP, and bound proteins were eluted with wash buffer containing 50 mM reduced glutathione. Following overnight TEV digestion at 4 °C, both MBP and GST elution were subjected to Strep-Tactin Sepharose resin, washed with wash buffer and eluted with wash buffer supplemented with 10 mM desthiobiotin.

For the DDD -Ube2e2-COP1-c-Jun-STK40 complex, untagged DDB1 and DDA1, TSF-tagged human DET1 and STK40, N-terminal GST- and C-terminal His-tagged COP1, MBP-tagged Ube2e2 and c-Jun were co-transfected into Expi293F cells. For the DDD-Ube2e2-COP1-ETS2 complex, untagged DDB1 and DDA1, TSF-tagged human DET1, N-terminal GST- and C-terminal His-tagged COP1, MBP-tagged Ube2e2 and ETS2 were used for co-transfection. Both complexes were purified using the same protocol as described for the DDD-Ube2e2-COP1-c-Jun complex. For the DDD-Ube2e2-COP1-ETS2 and DDD-Ube2e2-COP1-c-Jun-STK40 complexes, the corresponding plasmids were co-transfected into Expi293F cells. 12 h post-transfection, 1 mM sodium butyrate was added. After an additional 36 h, 10 μM MG132 was added for a further 12 h.

Cullin4A-RBX1-DDB1 complex (GST-Cullin4A, TSF-RBX1 and TSF-DDB1), neddylated Cullin4A-RBX1-DDB1 complex (Uba3, UBC12, APPBP1, HA-NEDD8, GST-Cullin4A, TSF-RBX1 and TSF-DDB1), COP1-c-Jun complex (GST-COP1-TSF and MBP-c-Jun), COP1-DET1-c-Jun complex (GST-COP1, TSF-DET1 and MBP-c-Jun) were co-transfected in Expi293F cells and purified in the same procedures as described above.

Uba1 was PCR-amplified from cDNA and cloned into pGEX-4T1 vector, which introduced an N-terminal GST tag followed by a TEV cleavage site and a C-terminal His tag. The construct was transformed into *E.coli* Rosetta cells (AlpaLifeBio, KTSM106L) for protein expression. Cells were grown in LB medium at 37 °C to an OD$_{600}$ of 0.5-0.6, then induced with 0.5 mM isopropyl-1-thio-β-d-galactopyranoside (IPTG) at 16 °C for 16–18 h. Harvested cells were resuspended in lysis buffer (20 mM HEPES-NaOH pH 7.4, 500 mM NaCl, 2 mM MgCl$_2$, 0.5 mM TCEP, 1 mM PMSF), then lysed by sonication. Cell debris were removed by centrifugation, and the supernatant was incubated with preequilibrated GST beads at 4 °C for 1.5–2 h. The beads were washed with wash buffer (20 mM HEPES-NaOH pH 7.4, 200 mM NaCl, 2 mM MgCl$_2$, 0.5 mM TCEP). Bound proteins were eluted using the same buffer containing 50 mM reduced glutathione. After TEV digestion overnight at 4 °C, the GST elution was then subjected to Ni-NTA (QIAGEN), washed with wash buffer and eluted with the same buffer containing 300 mM imidazole. The Ni elution was further loaded on a Superdex 200 Increased 10/300 GL column (Cytiva) preequilibrated with wash buffer. The purified Uba1 protein was concentrated and stored at −80 °C.

Ubiquitin was cloned into pGEX-4T1 vector for expression in *E.coil* Rosetta cells. Transformed cells were cultured in LB medium at 37 °C until reaching an OD$_{600}$ of 0.5–0.6, then induced with 0.5 mM IPTG at 16 °C for 16–18 h. Cells were harvested and resuspended in lysis buffer, lysed by sonication, and clarified by centrifugation. The supernatant was incubated with GST beads at 4 °C for 1.5–2 h. The beads were washed with wash buffer and the bound proteins were eluted with wash buffer containing 50 mM reduced glutathione. After overnight TEV digestion at 4 °C, the GST elution was then loaded on a Superdex 75 Increased 10/300 GL column (Cytiva) preequilibrated with wash buffer. The purified protein was concentrated and stored at −80 °C.

For Ube2d3-His$_6$, the gene was cloned into pGEX-4T1 for expression as an N-terminal GST fusion with a TEV cleavage site. The protein was purified in a similar procedure as described above. Ube2e2 and its mutants were purified following the same protocol as for Ube2d3-His$_6$.

## Cryo-EM grid preparation and data acquisition

The purified DDB1-DDA1-DET1 complex was prepared at a concentration of 0.2 mg/ml and applied to glow-discharged UltrAuFoil (1.2/1.3, 300 mesh) grid, with an incubation time of 10 sec. A total of 6,111 movies were collected at 300 kV with a Gatan Quantum energy filter (operated at 20 eV slit width) using a Gatan K3 direct electron detector. Data were acquired in counting mode at a defocus range of −1.2 μm to −1.8 μm, yielding a pixel size of 0.85 Å at the specimen level. The total electron dose was 52.6 e$^-$/Å$^2$ over a total exposure time of 2.497 sec, fractionated into 50 frames.

For the DDB1-DDA1-DET1-COP1 complex, the purified protein at concentrations of 0.535 or 0.3 mg/ml was applied to freshly glow-discharged UltrAuFoil (1.2/1.3, 300 mesh) grids for multiple collection sessions. The samples were vitrified after blotting for 3 sec using a Vitrobot Mark IV (FEI) with either 10 or 15 sec of incubation, blot force 0, at 100 % humidity, and 4 °C. A total of 23,850 movies were collected with a Titan Krios electron microscope (Thermo Fisher Scientific) operating at 300 kV with a Gatan Quantum energy filter (operated at 20 eV slit width) using a Gatan K3 direct electron detector. Data were acquired in counting mode at a defocus range of −1.0 μm to −1.8 μm, yielding a pixel size of 0.85 Å on the specimen level. The total electron dose on the specimen was 46.55, 46.32, 47.42, and 48.6 e$^-$/Å$^2$, with total exposure time of 2.497 or 1.997 sec, fractionated over 50 frames.

For the DDB1-DDA1-DET1-COP1-Ube2e2(S76G-C139S)-c-Jun complex, the purified protein was prepared at a concentration of 0.225 mg/ml and applied to glow-discharged UltrAuFoil (1.2/1.3, 300 mesh) grid, with an incubation time of 10 sec at 4 °C and 100 % humidity. A total of 5,543 movies were collected on a Titan Krios electron microscope operating at 300 kV equipped with Gatan K3 camera. Data were acquired in counting mode at a defocus range of −1.2 μm to −1.8 μm, corresponding to a pixel size of 0.85Å. Each movie consisted of 50 frames, with a total dose of 48.89 e$^-$/Å$^2$.

For the DDD-Ube2e2-COP1-c-Jun-STK40 complex, the purified protein was prepared at a concentration of 0.52 mg/ml and mixed with 0.02% N-Octyl-β-D-glucopyranoside (OG) prior to cryo-EM grids preparation to mitigate the effects of preferred orientation of the particles. A 4 μl of sample was loaded onto a glow-discharged UltrAuFoil 300 mesh R1.2/1.3 grids using a FEI Vitrobot with an incubation time of 12 sec at 4 °C and 100 % humidity. The grids were transferred to FEI Titan Krios electron microscope operating at 300 kV, and movies (32 frames, total accumulated dose 50 e$^-$/Å$^2$) were collected using a direct electron detector Gatan K3 in the counting mode with a defocus range from −1.5 to −2.3 μm. Automated single-particle data acquisition was performed with the EPU (Thermo Fisher Scientific) software at a nominal magnification of 105,000 ×, yielding a final pixel size of 0.83 Å.

## Negative stain EM imaging

A 4 μl aliquot of purified protein (about 0.02 mg/ml) was applied to a freshly glow-discharged carbon-coated 300 mesh grid (X-Pivot, XP-CF300) and incubated for 3 min. The grids were subsequently blotted with filter paper and negatively stained with 2% (w/v) uranyl acetate for 40 sec, after which excess liquid was removed with filter paper. Imaging was performed on a Talos 120 C transmission electron microscope (Thermo Fisher) performed at 120 kV in low-dose mode and imaged with a Ceta CMOS camera (Thermo Fisher). Data were collected at a nominal magnification of 73,000 × or 57,000 ×, corresponding to a pixel size of 1.91 Å or 2.45 Å, respectively.

## Cryo-EM data processing

The movies were initially aligned for all the datasets using MotionCor2 implementation in Relion 3 to correct specimen movement and then imported into cryoSPARC v4[58–60]. CTF (contrast transfer function) fitting and estimation were performed using patch CTF estimation. The blob picker and template picker were applied to pick the particles before undergoing 2D classification.

**DDB1-DDA1-DET1 complex.** A total of 6,111 movies were collected for the DDB1-DDA1-DET1 complex. Initial particle picking from 4.8 million particles was performed using a blob picker, followed by iterative cycles of 2D classifications. The 2.9 million particles were subjected to ab initio reconstruction to generate three classes. Following heterogeneous refinement, the best class was selected for non-uniform refinement, and a map at 2.7 Å was obtained from 1 million particles. After particle subtraction on the DDB1$^{BPB}$ domain and local refinement, the final map was estimated at 2.65 Å. The final map was postprocessed by DeepEMhancer[61].

**DDB1-DDA1-DET1-COP1 complex.** In total, 23,850 movies were collected for the DDB1-DDA1-DET1-COP1 complex, and 6,327 movies of all were collected by tilting 10 degrees to mitigate the preferred orientation of the particles. After initial 2D classification, good classes were selected as templates, combined with particles from 23,406 micrographs, and further subjected to 2D classification. Meanwhile, these classes were subjected to ab initio reconstruction. After rearrangement and removal of duplicates, particles were divided into three classes based on size. 66,513 particles in the first class were subjected to nonuniform refinement (C2 symmetry) and produced two layers after applying a mask and local refinement. The final map of layer 1 and layer 2 were estimated at 4.33 Å and 4.23 Å, respectively. The second class used non-uniform refinement (C2 symmetry) and heterogeneous refinement, resulting in 3.03 Å resolution reconstruction from 61,817 particles. 1.78 million particles in the third class were cleaned up by heterogeneous refinement and then further subjected to 3D classification. After nonuniform refinement, two good classes were divided and combined to do particle subtraction on the DDB1$^{BPB}$ domain. After local refinement, the final maps obtained from 401,370 or 705,861 particles were estimated at 2.93 Å and 2.92 Å, respectively.

**DDB1-DDA1-DET1-COP1-Ube2e2$^{S76G-C139S}$-c-Jun.** 5,543 movies were collected for the DDB1-DDA1-DET1-COP1-Ube2e2$^{S76G-C139S}$-c-Jun complex. After several rounds of cleaning of 2D classifications, three good classes were to be selected for the template picker according to different diameters. 1.36 million particles in the 300 Å class, 4.4 million in the 150 Å class, and 7.1 million in the 100 Å class were further subjected to 2D classifications. After removing duplicates, some particles were selected and subjected to ab initio reconstruction and nonuniform refinement. Subsequently, 42,109 particles were estimated at 3.62 Å. Simultaneously, 1.5 million particles were used for nonuniform refinement and heterogeneous refinement, resulting in 3 classes. 1,019,651 particles from two classes were combined and subjected to particle subtraction on the DDB1$^{BPB}$ domain, resulting in 3.26 Å resolution reconstruction after local refinement. 494,668 particles from another class were used in similar processing pipelines, leading to 3.25 Å resolution reconstruction.

**DDB1-DDA1-DET1-COP1-Ube2e2-c-Jun-STK40.** A total of 23,615 movies were collected for the DDB1-DDA1-DET1-COP1-Ube2e2-c-Jun-STK40 complex. The micrographs were subjected for blob pickers with varying particle diameters, and the extracted particles were pooled for interactives 2D classification. After removing duplicate particles and further cleaning through additional rounds of 2D classification, a subset of 274,513 dimeric-shaped particles was selected. These particles were subjected to non-uniform refinement, resulting in a reconstruct at 3.7 Å resolution.

The cryo-EM maps were post-processed using DeepEMhancer (tight target modes) for model building and figure preparation. All the reported resolutions were based on the gold standard FSC 0.143 criterion in Supplementary Fig. 13. We acknowledge that because of the orientation bias, the resolution in the least resolved direction is lower than the global average. The anisotropy of each reconstruction was evaluated to account for preferred orientation bias. Using the

Orientation Diagnosis tool in cryoSPARC, we calculated the conical FSC area ratio (cFAR) score, which quantifies directional resolution, and the sampling compensation factor (SCF)[62,63]. These data, providing a full account of the signal anisotropy for each map, are presented in Supplementary Table 6.

## Atomic model building and refinement

Since the datasets exhibit a preferred orientation problem that could compromise the quality and accuracy of the 3D reconstructions, we have complemented the structure analysis with the computational predictions to mitigate this limitation. Model building was achieved using AlphaFold2 predictions and previously reported structures as initial models. The coordinates of individual subunits from the following previously reported structures were rigid-body fitted into the cryo-EM reconstruction with UCSF chimera: DDB1-DDA1 (PDB ID: 6DSZ), Ube2e2 (PDB ID: 6W9A), DET1 (Alphafold model: AF-Q7L5Y6-F1; Uniprot: Q7L5Y6) and COP1 (Alphafold model: AF-Q8NHY2-F1; Uniprot: Q8NHY2). The models were subsequently rebuilt manually using Coot 0.8.9.3-pre. For all the models, regions exhibiting low Cα backbone density were removed, and in areas of high flexibility or low resolution, side chains were trimmed while the Cα connectivity and amino acid assignments were retained. Density corresponding to the endogenous Ube2e family member in the DDD-COP1 dataset was modelled as a polyalanine version of Ube2e2 with side chains removed. The coiled-coil of COP1 in stacked structures are restricted to only backbone fitting. Atomic coordinates were refined against unsharpened cryo-EM maps through an iterative process using Phenix (version 1.20.1–4487) real-space refinement, manual inspection and correction in Coot[64–66]. To prevent overfitting, the map weight was set to, 1 and secondary structure restraints were applied during real-space refinement. Model quality was assessed using MolProbity and map-model FSC[67]. Figures were prepared using UCSF Chimera version 1.15 or ChimeraX 1.4[68,69]. To analyze the intermolecular contacts, the predicted model of low resolution regions was superimposed onto the experimental structure and validated experimentally. The cryo-EM density maps have been deposited in the Electron Microscopy Data Bank (EMDB) under accession codes EMD-63371, EMD-63372, EMD-63374, EMD-63375, EMD-63397, EMD-63383, EMD-63385, EMD-63386, EMD-63565 and EMD-65758. The corresponding coordinates were deposited in the Protein Data Bank (PDB) under accession numbers 9LTJ, 9LTL, 9LTO, 9LTR, 9LUL, 9LTW, 9LTZ, 9LU1, 9M0Y and 9W90.

## Pull-down experiment

Pull-down experiments were performed in Expi293F cells co-transfected with plasmids as indicated in the figures. After 12 h of transfection, 1 mM sodium butyrate was added, followed by an additional 48 h of incubation. The cells were harvested 3 days after transfection and washed with ice-cold 1× PBS. The pellets were lysed in lysis buffer with protease inhibitors. Following centrifugation, the supernatant was incubated with 50 µl Strep-Tactin Sepharose resin, MBP resin, or GST beads (Beyotime, P2253) at 4 °C for 1 h. Beads were washed three times with lysis buffer supplemented with 2 mM ATP and eluted in wash buffer containing either 10 mM desthiobiotin, 20 mM maltose, or 50 mM reduced glutathione, as appropriate. The eluents were analyzed by SDS-PAGE or western blotting. All experiments were performed at least three times independently.

## In vitro ubiquitination assay

The ubiquitination assays were performed in 30 µl reaction volume containing 0.45 µM E1 (Uba1), 45 µM ubiquitin, 5 mM ATP, 4.5 µM Ube2d3 or Ube2e2 (WT, C139S or C139A), 1 µM CULLIN4-RBX1-DDB1 or neddylated-CULLIN4-RBX1-DDB1 complex, and 1 µM COP1-c-Jun or DET1-COP1-c-Jun or DDD-Ube2e2-COP1-c-Jun complex. Reactions were carried out in assay buffer containing 50 mM Tris-HCl pH 7.5, 50 mM NaCl, 10 mM MgCl₂, 2 mM DTT. Reaction mixtures were incubated for

2 h at 37 °C, then terminated by the addition of DTT-containing SDS sample buffer, followed by immunoblot analysis. All experiments were performed at least three times independently.

## LC-MS/MS Identification of unknown proteins co-purifying with the DDB1-DDA1-DET1-COP1 complex

Identification of unknown protein was preformed using an automated system that integrated rapid pepsin digestion, on-line desalting and high-resolution LC/MS/MS. Purified protein complexes were acidified and digested on a home-made immobilized pepsin column. Eluted peptides were desalted using trap column (1 mm × 15 mm, Acclaim PepMap300 C18, 5 µm, Thermo Fisher Scientific) for 5 min at a flowrate of 200 µL/min and 0.1% formic acid as mobile phase. Subsequent peptide separation was performed on the ACQUITY BEH C18 (2.1 × 50 mm) analytical column using a first gradient ranging from 9 to 45% of buffer B (80% acetonitrile and 0.1% formic acid) for 10 min followed by a second gradient ranging from 45 to 99% of buffer B for 1 min, at an overall flow rate of 50 µL/min. Peptides were ionized via electrospray ionization and analyzed by Orbitrap Eclipse (Thermo Fisher) mass spectrometer using data dependent tandem MS/MS acquisition. All MS/MS spectra were searched against the human uniport database using the Proteome Discoverer 2.5 software. The following parameters were used: No-Enzyme (Unspecific); MS tolerance, 10 ppm; MS2 tolerance, 0.02 Da; variable modifications, oxidation methionine (+15.995 Da). A maximum of 3 dynamic modifications was allowed per peptide. Peptide-spectral matches (PSMs) were validated using Percolator with a 1% false discovery rate (FDR) cut-off.

## Crosslinking mass spectrometry (XL-MS)

The final concentration of 0.5 µM DDD-Ube2e2-COP1-c-Jun was incubated with a final concentration of 0.1 mM DSBU crosslinker (Thermo Fisher, A35459) in the wash buffer (20 mM HEPES-NaOH pH 7.4, 200 mM NaCl, 2 mM MgCl₂, 0.5 mM TCEP) for 40 min at 25 °C and quenched with 10 mM Tris-HCl pH 8.0 for 10 min at 25 °C. Proteins were denatured with 10 mM DTT and 8 M urea for 60 min, and then alkylated with 50 mM Iodoacetamide (IAA) for 30 min in the dark. The protein samples were digested with trypsin (trypsin: protein = 1:20, w/w) at 37 °C. Tryptic peptides were desalted using C18 Stage Tips and speedvac the sample till dryness and store at −20 °C for LC-MS analysis. Mass Spectrometry analysis was performed with an Orbitrap Fusion mass spectrometer (Thermo Fisher Scientific) with a nanospray ion source and an EASY-nLC 1000 system (Thermo Fisher Scientific). The peptides were directly loaded to an analytical column (100 µm i.d × 20 cm) packed with 1.9 µm and 120 Å ReproSil-Pur C18 resins (Dr. Maisch GmbH, Ammerbuch, Germany). A binary mobile phase system was used, buffer A: 0.1% (v/v) formic acid (FA) in water and buffer B: 0.1% (v/v) FA in acetonitrile (ACN). Peptides were eluted with a gradient of 7%-22% buffer B for 50 min followed by 22%-35% buffer B for 10 min. The full mass scan was acquired from m/z 350 to 1550 with resolution 120,000 at a target of 2e5 ions, and the MS/MS spectra were obtained in a data dependent acquisition in the top speed mode with higher-energy collision dissociation (target 5e4 ions, max ion injection time 100 ms, isolation window 1.6 m/z, normalized collision energy 25%). The dynamic exclusion time was set to 30 s. Precursor ions with unassigned charge state as well as charge state of 1+ or superior to 6+ were excluded from fragmentation. Cross-linked peptides were analyzed and identified using Proteome Discoverer 2.4 and XlinkX software 2.4 (Thermo Fisher). The following settings were applied: two missed cleavage sites for trypsin per peptide were allowed, with cysteine car-bamidomethylation as a fixed modification, with methionine oxidation and lysine DSBU Hydrolyzed as dynamic modification. Searches were performed against a database including the sequences of DDB1, DDA1, DET1, COP1, Ube2e2 and c-Jun. The search results were filtered based on precursor tolerance (±10 ppm) and fragment tolerance (±0.02 Da). The FDR threshold was set to 1% at the crosslink and CSM levels.

**Reporting summary**

Further information on research design is available in the Nature Portfolio Reporting Summary linked to this article.

## Data availability

The Cryo-EM density maps and associated masks have been deposited in the EMDB under accession number EMD-63371, EMD-63372, EMD-63374, EMD-63375, EMD-63397, EMD-63383, EMD-63385, and EMD-63386, EMD-63565, and EMD-65758. Atomic coordinates were deposited in the PDB with accession number 9LTJ, 9LTL, 9LTO, 9LTR, 9LUL, 9LTW, 9LTZ, 9LU1, 9M0Y, and 9W90. The mass spectrometry proteomics data were deposited to the ProteomeXchange Consortium through the PRIDE partner repository with the dataset identifier PXD071633[70]. Source data are provided with this paper.

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

## Acknowledgements

M.-Y.S. is an investigator of SUSTech Institute for Biological Electron Microscopy. We thank Shuyun Tian for the assistance in the early stage of the project. The authors thank the Cryo-Electron Microscopy Center (SUSTech), cryo-EM (KEMC) and advanced mass spectrometry facility (KMS) of the Kobilka Institute of Innovative Drug Discovery (CUHKSZ) and SUSTech Core Research Facilities for technical assistance. This work was supported by the National Natural Science Foundation of China (32571412 to M.-Y.S.), Shenzhen Medical Research Fund (Grant No. B2402014 to G.S. and B2502011 to M.-Y.S.), Research Fund For Inter-national Scientists of National Natural Science Foundation of China (W2532021 to G.S.), Shenzhen Science and Technology Program (20231120103446003 to M.-Y.S.), Medical Research Innovation Project G030410001 and start-up funding from SUSTech (to M.-Y.S.).

## Author contributions

M.-Y.S. initiated the project, designed and performed the biochemical and structural experiments. S.W. performed the biochemical experi-ments; F.T. was responsible for cryo-EM data screening and data col-lection; G.S. contributed to experimental design and manuscript writing; F.R. contributed to reagents and discussion; M.-Y.S. conceived the project, designed the study, and wrote the original draft with input from all authors.

## Competing interests

The authors declare no competing interests.
