## [Transparent Peer Review file · Nature Communications]

Cryo-EM structure of the human COP1-DET1 ubiquitin ligase complex

Corresponding Author: Dr Ming-Yuan Su

Version 0:

Reviewer comments:

Reviewer #1

(Remarks to the Author)

The CULLIN4-COP1/DET1 complex plays a crucial role in targeting several transcription factors for degradation in mammals. However, its structural assembly has remained poorly understood. This study represents a significant advancement in addressing this knowledge gap. Utilizing cryo-electron microscopy, the authors reveal the structures of human COP1 in complex with DDB1-DDA1-DET1 (DDD) and UBE2E2, identifying multiple distinct assembly states. They demonstrate that DET1 acts as a structural scaffold linking COP1 to DDD/UBE2Es, while DDB1 recruits the CULLIN4-RBX1 complex, which subsequently engages additional E2 enzymes (E2D3) via RBX1 to facilitate ubiquitin chain elongation. The study underscores the dynamic interplay between different structural states of the CULLIN4-COP1/DET1 E3 ligase complex and its activation through substrate binding, providing valuable insights into substrate ubiquitination and the higher-order assembly of E3 ligases.

While this work represents a substantial contribution, several aspects warrant further clarification:

1. Direct Interaction Between COP1 and UBE2Es

The structural data in Figure 3 suggest that COP1 and UBE2Es do not interact in the absence of DET1. However, once DET1 brings them into proximity, do they establish a direct interaction? The authors compare the COP1-UBE2E2 complex to the TRIM21-UBE2E1 complex, implying a conventional RING-UBC domain interaction. To test this, could the authors introduce mutations in the RING loop residues predicted to mediate E2E2 UBC binding and assess whether these mutations disrupt the interaction and higher-order complex assembly?

2. Catalytic Activity of E2E2 in Complex Assembly

Is the catalytic activity of E2E2 required for the formation of the COP1/DDD complex? In their in vitro ubiquitination assays, the authors could test:

o COP1-JUN with E2E2 alone, including a catalytically inactive E2E2.

o COP1-JUN with DET1-E2E2, excluding DDB1/DDA1/CUL4/RBX1, to determine whether JUN undergoes ubiquitination.

These experiments would clarify whether DET1-E2E2-COP1 dimers can function independently as an E3 ligase complex in cells. In other words, is DET1-E2E2-2xCOP1 sufficient to recruit substrates and catalyze ubiquitination?

3. Biological Relevance of the Stacked Filament Structure

The observed stacked filament arrangement is intriguing. Could the authors comment on its potential physiological relevance? Might it represent an inactive state, an artifact of sample preparation, or a structure that only forms under specific conditions?

4. Potential Functional Role of the Stacked State

Beyond being an inactive complex, could this stacked conformation serve as a platform to sequester multiple substrate copies or substrates with adaptors, such as Tribbles and STK40? Given that some substrates require adaptors for COP1-mediated ubiquitination (e.g., Trib1 for C/EBP α [PMID: 23884858] or STK40 for JUN [PMCID: PMC6597834]), could the 'active' structure accommodate both an adaptor and a substrate simultaneously via two COP1 molecules (as shown in Figure 4)? Have the authors attempted to model STK40 into their structure or obtain a COP1/DDD-E2E2/STK40/JUN complex? Perhaps stabilizing cJUN in the presence of an adaptor (and a proteasome inhibitor) could provide additional insights, as JUN was not observed in the reconstruction.

5. Structural Basis for the Inactive-to-Active Transition

The authors propose a phosphorylation-dependent activation mechanism, referencing the known inactivation of the CULLIN4-COP1/DET1 complex via MAPK-mediated DET1 phosphorylation (Zhang et al., PNAS, 2017). Did their structural data provide any insights into this regulatory mechanism?

6. Clarification of COP1 Dimerization

The conclusion that COP1 dimerizes via its coiled-coil domain (Figure 1) appears inconsistent, as the authors later state that these domains were unresolved (page 6). Could they clarify this point?

7. Comparison with Recent Literature

It is crucial to assess whether the proposed structural dynamics of DET1 and the assembly of DDD/UBE2Es align with recent findings by Burgess et al. (Science Advances, 2025).

Addressing these questions would further solidify the mechanistic insights presented in this important study.

Reviewer #2

(Remarks to the Author)

The authors present novel and very interesting data on DET1 -COP1 complexes in humans. Several degrees of structural complexity are provided, association with different Ube2e proteins are described and an interesting hypothesis for substrate association and complex dissociation is proposed. In the overall the structural data is novel and opens the door to new hypothesis and possibilities in the field. However, I have some concerns given below.

- 1) The work by Burgess et al, has been uploaded in bioRxiv in June 2024 and recently published in Science Advances. However, it is not being properly referenced in the text and doesn't appear in the reference list. It should also be mentioned in introduction and properly discussed. The authors could also mention on what they advance the knowledge regarding the data in Burgess et al., 2025, Science Advances. For instance, that for the first time a DET1-COP1 interaction and the DDD complex+COP1 is being seen and doesn't rely in prediction, besides all the supra-molecular complex structures described.
 - 2) Most of the molecular structures are lacking biochemical validation, and artefact formation should be discarded. Interactions with protein mutant variants should be performed to validate the structural data as the dimeric/mirror DDD and the stacking structures. Perhaps gel filtration assays with these mutated protein versions might also help.
 - 3) Additionally, despite COP1-c-Jun interaction was biochemically explored, there is no additional biochemical evidence that the association with c-Jun triggers the complex dissociation.
 - 4) Could authors experimentally demonstrate that dimeric DDD-COP1 complex is inactive whereas the monomeric DDD-COP1 complex is active?
 - 5) c-Jun doesn't appear in the images captured with the DDD complex. What is the evidence that the protein is still there and intact when the images are collected?
 - 6) It is not fully understandable how the authors can extract the conclusions described between lines 298 and line 307. First, there is little difference between lines 6 and 7 and these differences might be due to the fact that in line 7 there is a sum of the effects of lines 5 and 6. In addition, the two-step hypothesis described in line 305/306 in which DET1 collaborates with COP1 to recruit Ube2d3, would, in base of the experiment (Fig. 5), be unnecessary, since COP1 can promote c-Jun polyubiquitination without DET1. Then, the demonstration claimed in line 319 regarding the priming of ubiquitin in substrates lacks a solid basis.
 - 7) It is not fully understood why the E1 enzyme is not necessary for the first step (ubiquitin activation) in the in vitro ubiquitination assays shown in Figure 5.
- () The authors mention at the end of the Discussion (lines 363-364) "Future research should investigate the interplay of COP1 with a broader range of substrates to determine how structural transitions are fine-tuned for substrate specificity." This might include p53. Could the structural models solved by the authors help to determine whether whether p53 is a bona fide substrate of COP1 in vivo?

Minor comments:

Paragraph between lines 231 and 252 cuts a main idea in the text and could perhaps be moved to another position.

Line 76. CRLs should be corrected for CRL4s.

Line 84 – The provided references do not provide evidence for DDA1 participation in the CDDD complex in plants neither of its involvement in photomorphogenesis.

Line 99 – The correct reference for COP1-DET1 interaction in plants is Cañibano et al., 2021, Molecular Plant

Line 142 – Which previous studies? References are missing. The use of references could be checked across the work.

Reviewer #3

(Remarks to the Author)

Reviewer #4

(Remarks to the Author)

This paper presents the cryo-EM structures of the human COP1-DET1 ubiquitin ligase complex, providing insights into the distinct assembly states of COP1 in complex with the DDB1-DDA1-DET1 (DDD) complex and a potential E2 enzyme. While the study provides valuable structural data, several critical issues need to be addressed before the paper can be accepted.

The primary concern lies in the structural data processing and modeling. The authors provide only two maps out of eight, which raises questions about the completeness and quality of the data. Despite claims of a 2.65 Å reconstruction for the DDD complex, the DET1 component is poorly resolved, even at lower thresholds. At higher thresholds, over 50% of the

potential is missing, and there are significant issues with the Ramachandran plot, which shows 12 outliers. These concerns are compounded by the fact that only partial data has been shared, and the maps presented appear suboptimal.

The authors mention using DeepEMhancer for post-processing the maps, but it is crucial to clarify where exactly these processed maps are shown in the manuscript. Since these maps are not fully derived from the raw data, they should not be deposited to the database. Instead, the raw maps, along with the sharpened post-processed versions, should be deposited for review.

Moreover, Figure S1, which shows the 2.65 Å map without DeepEMhancer, is of poor quality. In Figure S3, the local resolution scales for the middle and right structures are inconsistent. In Figure S4, the additional potential observed seems to fit any E2, not a specific one, and in Figure 2, the map in panel B does not support the modeling of the coiled coil. Given these issues, it is difficult to fully trust the accuracy of the structural determination, which appears to lack sufficient quality control.

Additional comments:

- Introduction: Line 77 refers to CRLs consisting of a scaffold subunit CULLIN4, but it is important to note that there are at least nine Cullin proteins in the CRL family. A summary paragraph at the end of the introduction, such as “In this study, we used single-particle cryo-EM to...”, would help clarify the study’s aims and approach.
- Figures and Structural Models: Figure 1B should be cited earlier when discussing the domains of DDB1. The authors present many structural models but fail to provide sufficient cryo-EM maps to support these models, which weakens the validity of the claims.
- Line 191: The statement about C3 symmetry is inaccurate. A revision is needed.
- Line 202: This paragraph is unclear. It is unnecessary to superimpose models to predict zinc fingers, as AlphaFold typically provides accurate predictions. If AlphaFold models were used, this should be stated clearly.
- Line 223: The content of this paragraph should be moved to the discussion, as it feels out of place in the results section.
- Figure 3D: The map quality does not appear to support side chain assignments, and this should be addressed.
- Line 247: The authors justify mutating to glycines instead of alanine. Glycine mutations can drastically alter protein structure, so this decision should be more thoroughly explained.
- Figure 5: An anti-ubiquitin blot should be included to strengthen the findings, as it is critical for validating the ubiquitination results.

While the study offers important structural insights into the COP1-DET1 complex, the concerns regarding the quality and completeness of the cryo-EM data and the corresponding structural modeling are significant. The authors need to provide a more rigorous analysis, including the full set of maps and improved structural refinements, to support their conclusions. I recommend addressing these issues and resubmitting the paper once the data processing and modeling have been adequately corrected.

Version 1:

Reviewer comments:

Reviewer #1

(Remarks to the Author)

My concerns have been addressed.

Reviewer #4

(Remarks to the Author)

The authors have made a genuine effort to improve the clarity and presentation of the manuscript in response to my previous comments. The revised version is notably clearer in both the writing and the organization. However, I still have serious concerns regarding the quality and reliability of the structural data, which directly impact the central claims of the manuscript. Unfortunately, the cryo-EM map quality remains insufficient to support the modeling and interpretations made in the paper.

Below are specific concerns for each reported structure:

1. 9LTJ: A significant portion of DET1 is poorly resolved. The authors used an extremely low threshold value in the EM validation server, making the map appear to envelop nearly all atoms — a misleading practice that inflates interpretability. The reported resolution is inconsistent (3.25 Å in the validation report vs. 2.65 Å in the manuscript). Moreover, the model built on the 2.65 Å map contains 3.28% rotamer outliers and Ramachandran violations, which is unacceptable for such a resolution claim.
2. 9LTW: The sharpened map is visually almost identical to 9LTJ. The unsharpened map shows some additional density, but it is far too ambiguous to support confident placement of anything.
3. 9LTZ: The map appears streaky and lacks interpretable features. Manual modeling would not be feasible without AlphaFold models, calling into question the legitimacy of the model building.
4. 9LU1: This is the only map in the set that is of generally acceptable quality. However, it still barely supports the modeling in Fig. 4.
5. DDD-ube2e2-COP1-c-Jun-STK40: It is unclear which PDB entry this corresponds to. The validation report provided shows “NOT for manuscript review”, and again, the thresholding appears artificially low. The map is extremely streaky and the backbone is not traceable (also impossible for the deep-enhanced model). This pattern is suggestive of severe preferred orientation.

6. 9LTO: Although the additional density is less streaky than others, the placement of an E2 is still speculative at best, with no clear features supporting its orientation or conformation.
7. 9LTL: This map does not allow tracing of the main chain, yet the authors report a resolution of 2.93 Å — a value that is clearly inconsistent with the visible density.
8. 9LTR: The map is again streaky, likely due to preferred orientation and inclusion of low-quality particles. The reported resolution of 3.03 Å appears greatly exaggerated.
9. 9LUL, 9M0Y: While the coiled-coils are faintly visible, there is no meaningful detail beyond the global shape. The reported resolutions in the 4.x Å range do not reflect the poor interpretability of these maps.
10. Supplemental Fig. S9. The FSC plots lack numerical resolution values, which limits their interpretability. In panels A, B, and D–G, more than half of the frequency components are zero, suggesting that unnecessary large boxes were used for the analysis. Additionally, in panel C, the FSC curve does not drop to zero at high frequency, which raises concerns about potential particle duplication or overfitting.
11. General Recommendation: I strongly recommend that the authors seek guidance from an experienced cryo-EM structural biologist regarding data processing, map validation, and model building. The quality of both maps and models is foundational to structural biology, and in its current form, the manuscript does not meet the necessary standards for publication.

Despite improved clarity in writing, the structural data remain insufficient in quality and reliability. Therefore, I cannot support publication of this work in its current form.

Version 2:

Reviewer comments:

Reviewer #4

(Remarks to the Author)

Thank you for providing additional explanations and clarifications in response to my previous comments. At this stage, I do not have further constructive suggestions to offer. However, I remain concerned about the overall quality and interpretability of several cryo-EM maps and the degree to which the structural models rely on AlphaFold predictions. While I appreciate the effort made in this revision, these issues continue to limit my confidence in the structural conclusions.

We sincerely thank the reviewers for their insightful feedback and constructive suggestions to further improve the manuscript. Below is a point-by-point response to all the comments.

Reviewer #1 (Remarks to the Author):

The CULLIN4-COP1/DET1 complex plays a crucial role in targeting several transcription factors for degradation in mammals. However, its structural assembly has remained poorly understood. This study represents a significant advancement in addressing this knowledge gap. Utilizing cryo-electron microscopy, the authors reveal the structures of human COP1 in complex with DDB1-DDA1-DET1 (DDD) and UBE2E2, identifying multiple distinct assembly states. They demonstrate that DET1 acts as a structural scaffold linking COP1 to DDD/UBE2Es, while DDB1 recruits the CULLIN4-RBX1 complex, which subsequently engages additional E2 enzymes (E2D3) via RBX1 to facilitate ubiquitin chain elongation. The study underscores the dynamic interplay between different structural states of the CULLIN4-COP1/DET1 E3 ligase complex and its activation through substrate binding, providing valuable insights into substrate ubiquitination and the higher-order assembly of E3 ligases.

Dear Reviewer #1,

We are grateful for your thoughtful review and for the positive evaluation regarding our work.

While this work represents a substantial contribution, several aspects warrant further clarification:

1. Direct Interaction Between COP1 and UBE2Es

The structural data in Figure 3 suggest that COP1 and UBE2Es do not interact in the absence of DET1. However, once DET1 brings them into proximity, do they establish a direct interaction? The authors compare the COP1-UBE2E2 complex to the TRIM21-UBE2E1 complex, implying a conventional RING-UBC domain interaction. To test this, could the authors introduce mutations in the RING loop residues predicted to mediate E2E2 UBC binding and assess whether these mutations disrupt the interaction and higher-order complex assembly?

Thank you for this question. We engineered the DET1 mutants that would impact the binding with COP1 (R488A/H485G; R488A/H485G/R306G/S312G; R488A/H485G/E380A/T382A) and co-transfected them with COP1 and Ube2e2 (Figure R1A-B). In reciprocal pull-down assays, we observed that the reduction of COP1-DET1 interaction resulted in the decreased amount of Ube2e2 co-purification, suggesting that DET1-mediated proximity does not allow COP1 and Ube2e2 to establish the direct interaction (Figure R1C). Furthermore, guided by the structural comparison and sequence alignment of TRIM21^{RING} (PDB: 6FGA) and COP1 (Figure R1D), we also expressed a COP1^{RING} construct (residues N121-S210) and carried out the reciprocal pull-down assays with Ube2e2. The results indicated that there is no detectable interaction between them (Figure R1E). Together, these findings confirm DET1 is essential as a bridge and that COP1 may not form a canonical RING-E2 interaction with Ube2e2. This clarifies the unique assembly mode in this E3 ligase. We have included these new data and the corresponding figures into the revised manuscript (revised Figure 2 and Figure S6; lines

213-222 and 259-265).

Figure R1. DET1 acts as a bridge between COP1 and Ube2e2.

A-B, Close-up view of the molecular interface between DET1 and COP1^{WD40a} or COP1^{WD40b}. The key residues contributing to the interaction are labelled. **C**, Pull-down experiment of wild type or mutated TSF-DET1 in cells co-transfected with His-Ube2e2 and GST-COP1. Cell lysates were divided in two fractions and incubated with GST and strep beads. The eluents were analysed by SDS-PAGE and western blotting. WT, wild-type. TSF, twin-strep-FLAG tag. **D**, Sequence alignment of TRIM21^{RING} and COP1^{RING}. **E**, Pull-down experiment of GST-COP1^{RING} with MBP-Ube2e2. Cell lysates were divided in two fractions and incubated with GST and MBP beads. The eluents were analysed by SDS-PAGE. Data in **C** and **E** are representative of three independent experiments.

2. Catalytic Activity of E2E2 in Complex Assembly

Is the catalytic activity of E2E2 required for the formation of the COP1/DDD complex? In their *in vitro* ubiquitination assays, the authors could test:

- o COP1-JUN with E2E2 alone, including a catalytically inactive E2E2.
- o COP1-JUN with DET1-E2E2, excluding DDB1/DDA1/CUL4/RBX1, to determine whether JUN undergoes ubiquitination.

These experiments would clarify whether DET1-E2E2-COP1 dimers can function independently as an E3 ligase complex in cells. In other words, is DET1-E2E2-2xCOP1 sufficient to recruit substrates and catalyze ubiquitination?

We greatly appreciate this valuable suggestion which implement improvements to our assays. To directly address if DET1-Ube2e2-COP1 dimer is sufficient for substrate recruitment and ubiquitination, we performed the proposed *in vitro* ubiquitination assays (Figure R2). Here is the summary for the results:

- (1) C-Jun is not ubiquitinated by COP1 in the presence of Ube2e2 alone (wild type or catalytically inactive C139S or C139A, lanes 1-3);

- (2) COP1-c-Jun-DET1-Ube2e2 complexes, even with DDB1-DDA1 and CULLIN4-RBX1 (neddylated or not), do not catalyze c-Jun ubiquitination (lanes 4-7).
- (3) Efficient c-Jun polyubiquitination requires both neddylated CULLIN4-RBX1-DDB1-DDA1 and Ube2d3, in addition to COP1-DET1-Ube2e2.

Therefore, DET1-Ube2e2-COP1 alone is not sufficient for catalytic activity. These new data and the corresponding figure are now incorporated into the revised manuscript (revised Figure 5; lines 370-375).

Figure R2: *In vitro* ubiquitination assay of COP1-c-Jun complex or COP1-DET1-c-Jun complex. *In vitro* ubiquitin assay of COP1-c-Jun complex was incubated with E1 (Uba1), ubiquitin, ATP and Ube2d3, wild type or catalytic inactive Ube2e2 mutants shown on lanes 1-3. COP1-DET1-Ube2e2-c-Jun complex was incubated with E1 (Uba1), ubiquitin and CULLIN4-RBX1 or neddylated CULLIN4-RBX1 shown on lanes 4-7. Immunoblotting was performed with an antibody against c-Jun and ubiquitin. The presence (●) or absence (x) of components in the reaction are indicated. The experiment was repeated independently three times with similar results.

3. Biological Relevance of the Stacked Filament Structure

The observed stacked filament arrangement is intriguing. Could the authors comment on its potential physiological relevance? Might it represent an inactive state, an artifact of sample preparation, or a structure that only forms under specific conditions?

Thank you for highlighting this fascinating aspect. Our purifications from mammalian cells reveal both dimeric and higher-order (stacked) assemblies, suggesting these states can form

in a cellular context. In the DDD-COP1 preparations and upon co-expression of c-Jun or other substrates, we observed a shift towards the dimeric species. We therefore propose that the stacked assembly represents an autoinhibited, inactive state that may be sensitive to substrate abundance or other cellular cues, whereas the dimeric assembly corresponds to the active state. Notably, the dimeric observed in DDD-COP1 preparations likely reflects a fraction co-purified with endogenous substrates and exhibits the same architecture as the purified DDD-Ube2e2-COP1-c-Jun complex (supported by SEC profiles and negative stain EM; Figure R8). Consistently, substrate-containing fractions support robust c-Jun polyubiquitination *in vitro* (Figure R9). Together, these data support a model in which the stacked assembly is autoinhibited state and the dimer as the active architecture. Future studies-such as isolating endogenous complex from cells, in situ imaging, or systematic varying experimental parameters will further clarify the physiological relevance and regulation of these states.

4. Potential Functional Role of the Stacked State

Beyond being an inactive complex, could this stacked conformation serve as a platform to sequester multiple substrate copies or substrates with adaptors, such as Tribbles and STK40? Given that some substrates require adaptors for COP1-mediated ubiquitination (e.g., Trib1 for C/EBP α [PMID: 23884858] or STK40 for JUN [PMCID: PMC6597834]), could the 'active' structure accommodate both an adaptor and a substrate simultaneously via two COP1 molecules (as shown in Figure 4)? Have the authors attempted to model STK40 into their structure or obtain a COP1/DDD-E2E2/STK40/JUN complex? Perhaps stabilizing cJUN in the presence of an adaptor (and a proteasome inhibitor) could provide additional insights, as JUN was not observed in the reconstruction.

We greatly appreciate this intriguing prospect. Structurally, our data indicate that only the uppermost COP1 WD40 domains within the stacked filament are accessible to substrates, while the COP1 WD40 in the other layers are occluded. This arrangement is inconsistent with simultaneous multi-substrate/adaptor loading across layers, arguing against the stacked assembly serving as a platform for the simultaneous recruitment of multiple substrates or for adaptor-substrate complexes.

Regarding adaptors such as Tribbles or STK40, it is challenging to confidently model STK40 within our reported structure for the following reasons. Previous studies (PMID: 31092598, 28089446) demonstrated that the C-terminal region of STK40 is required for its interaction with both c-Jun and COP1, and that STK40 contains a VP motif responsible for COP1 WD40 binding. However, details of the interaction between STK40 and c-Jun remain unclear. To investigate this, we performed the co-expression and pull down experiments (Figure R3). GST-COP1 could pull down TSF-STK40, but no direct interaction was detected between TSF-STK40 and MBP-c-Jun. When all three plasmids were co-expressed, interaction were detected between them, indicating that COP1 bridges STK40 and c-Jun and/or engaging them simultaneously.

Figure R3: Co-expression and *in vitro* reciprocal pull down of GST-COP1 with TSF-STK40, TSF-STK40 with MBP-c-Jun, and GST-COP1 with MBP-c-Jun/TSF-STK40.

Cell lysates were divided into three fractions and incubated with GST, strep and MBP beads. The eluents were analysed by SDS-PAGE and western blotting.

To further investigate, we performed structural predictions using AlphaFold3 on the following assemblies and some of the results are shown below (Figure R4).

1. COP1 dimer with c-Jun (ipTM/pTM = 0.41/0.43)
2. COP1 dimer with c-Jun dimer (ipTM/pTM = 0.37/0.4)
3. COP1 dimer with STK40 (ipTM/pTM = 0.4/0.42)
4. COP1 dimer with STK40 and c-Jun (ipTM/pTM = 0.38/0.41)
5. COP1 dimer with c-Jun dimer and STK40 (ipTM/pTM = 0.36/0.4)
6. COP1 dimer with c-Jun dimer and STK40 dimer (ipTM/pTM = 0.38/0.44)

The AlphaFold3 predictions remain ambiguous, as reflected by the low ipTM/pTM scores, suggesting that the dynamic regions of these protein complexes are unlikely to be stabilized without additional factors. Notably, the predictions hint that STK40 engages COP1 dimer through a conserved short peptide (residues 344-351, containing the VP motif for COP1 binding). However, outside of this short binding region, STK40-c-Jun adopts highly variable conformations across different predictions, further underscoring their dynamic properties.

Figure R4: AlphaFold3 prediction of COP1 with STK40 and c-Jun.

A, The AlphaFold3 model of the STK40 and COP1 dimer. STK40 (residues 339-357) binds to the COP1^{WD40} domain. **B**, Sequence alignment of reported COP1 substrates and adaptors. The conserved VP motifs are shown in red. Close-up views of the interface between COP1^{WD40}: STK40 (model 1). **C**, The AlphaFold3 algorithm predicts the model of the STK40, c-Jun and COP1 dimer. STK40 and c-Jun bind to the WD40 domains of two COP1 molecules, respectively.

In line with your suggestion, we considered that STK40 might stabilize c-Jun, thus enabling its visualization in cryo-EM reconstructions experimentally. To test this, we co-transfected STK40 with our complex and collected cryo-EM data (Figure. R5A-C). The DDD-Ube2e2-COP1-c-Jun complex co-expressed with STK40 exhibited an overall structure similar to that without STK40, with an additional weak density localized to the COP1 dimerization region (Figure R5D). To further characterize this region, we generated a COP1 dimer model using AlphaFold3 prediction and superimposed it to our experimental map. The overall chimera structure displayed a good agreement with the 2D class averages, and the flexible regions of COP1 corresponded to the two protrusions observed in the 2D class averages (Figure R5I, red arrows). These results have been incorporated in revised Figure 4 and Figure S10; lines 335-353.

However, upon completing the processing pipeline, neither STK40 nor c-Jun yielded

interpretable density in the final reconstructions. We also repeated the expression with MG132 treatment, but observed no improvement in protein yield or visibility in cryo-EM.

Figure R5: Cryo-EM data processing for DDB1-DDA1-DET1-Ube2e2-COP1-c-Jun and STK40 complex.

A, Coomassie blue-stained SDS-PAGE analysis of the purified DDD-Ube2e2-COP1-c-Jun-STK40 complex. MW, molecular weight. TSF, twin-strep-FLAG. **B**, Representative cryo-EM micrograph of the DDD-Ube2e2-COP1-c-Jun-STK40 complex. Scale bar, 150 nm. **C**, Representative 2D class averages of the DDD-Ube2e2-COP1-c-Jun-STK40 complex. **D**, Flow chart of cryo-EM data processing. **E**, FSC plots between two independently refined half-maps with no mask (blue), loose mask (green),

tight mask (red), and corrected (purple). A cut-off of 0.143 (black line) was used to estimate the resolution. **F**, Angular particle distribution calculated in cryoSPARC for particle projections. The heatmap shows the number of particles for each viewing angle. **G**, Cryo-EM density map of DDD-Ube2e2-COP1-c-Jun-STK40 complex post-processed by DeepEMhancer. **H**, Local resolution map, colored-coded as indicated by the scale. **I**, The coordinate file of the DDD-Ube2e2-COP1-c-Jun-STK40 complex fitted into the cryo-EM map, presented from various views. The predicted COP1 dimer was further fitted into the experimental maps, and the resulting model shows the partial coiled-coil domain protruding in a manner similar to that observed in the 2D class averages. **J**, The COP1 dimer resolved from the DDD-Ube2e2-COP1-c-Jun and DDD-Ube2e2-COP1-c-Jun-STK40 experimental data is similar to the predicted model, which is further colored by predicted local distance difference test (pLDDT) and shown on the right.

In summary, we pursued several strategies:

- DDB1-DDA1-DET1-Ube2e2-c-Jun with MG132 treatment
- DDB1-DDA1-DET1-Ube2e2 with MBP-c-Jun (which facilitated c-Jun visualization)
- DDB1-DDA1-DET1-Ube2e2-c-Jun with STK40
- DDB1-DDA1-DET1-Ube2e2-c-Jun with STK40 and MG132 treatment

c-Jun could be detected by SDS-PAGE, but we consistently failed to obtain additional resolved density for it, or for STK40, in the cryo-EM reconstructions. Thus, under our experimental conditions, neither c-Jun nor its adaptor STK40 associates stably enough within the COP1 complex to permit structural resolution, likely due to inherent conformational flexibility, which is consistent to the structural predictions.

5. Structural Basis for the Inactive-to-Active Transition

The authors propose a phosphorylation-dependent activation mechanism, referencing the known inactivation of the CULLIN4-COP1/DET1 complex via MAPK-mediated DET1 phosphorylation (Zhang et al., PNAS, 2017). Did their structural data provide any insights into this regulatory mechanism?

Thank you for raising this regulatory question. In Zhang et al. (PNAS, 2017), they demonstrated that phosphorylation of DET1 Ser458 by MAPK inhibits CRL4^{COP1/DET1} by an as yet unknown mechanism. To determine whether DET1 S458 phosphorylation affects the complex assembly, we generated phosphomimic S458D and phosphor-dead S458A DET1 mutants and performed *in vitro* pull-down assays. Our results showed no disruption in the formation of the DDB1-DDA1-DET1-COP1 complex with either mutant (Figure R6A). We further repeated the protein co-expression and tandem purifications of the His-DDB1/DDA1/TSF-DET1^{S458D}/GST-COP1 to assess its overall conformation (Figure R6B-C). By negative stain electron microscopy, the DDD-COP1 complex containing the S458D phosphomimic mutant exhibited particles similar to those of the wild-type complex. In contrast, the DDD-Ube2e2-COP1-c-Jun (active state) complex showed distinctly different particles. These findings indicate that phosphorylation at DET1 Ser458 does not induce a major structural change or disassembly of the DDD-COP1 complex. Further studies will be required to elucidate the precise mechanism by which Ser458 phosphorylation regulates CRL4^{DET1-COP1}

activity. These new data and the corresponding figure are now incorporated in the revised manuscript (revised Figure S13 and lines 429-441).

Figure R6: Phosphomimic S458D of DET1 does not affect DET1 binding to COP1.

A, Pull-down experiment of wild type or S458A/S458D TSF-DET1 mutant in cells co-transfected with GST-COP1, His-DDB1, DDA1. Cell lysates were divided in two fractions and incubated with GST and strep beads. The eluents were analyzed by SDS-PAGE and western blotting. TSF, twin-strep-FLAG tag. **B**, Tandem affinity purification results of GST-COP1, His-DDB1, DDA1 and TSF-DET1^{S458D}. The target protein was obtained by GST affinity purification (*left*), TEV overnight digestion, and Strep affinity purification (*right*). **C**, The negative stain micrograph of DDB1-DDA1-DET1-COP1, DDD-Ube2e2-COP1-c-Jun and DDB1-DDA1-DET1^{S458D}-COP1 strep elution. Scale bar, 100 nm.

6. Clarification of COP1 Dimerization

The conclusion that COP1 dimerizes via its coiled-coil domain (Figure 1) appears inconsistent, as the authors later state that these domains were unresolved (page 6). Could they clarify this point?

Bianchi *et al.* previously demonstrated that human COP1 can dimerize via its coiled-coil by co-transfecting HA-tagged full length COP1 with either flag-tagged full length COP1 or various COP1 deletion constructs. Co-immunoprecipitation using anti-HA antibodies revealed that only constructs containing the coiled-coil can pull down HA tagged full length COP1 (PMID: 12615916), therefore we concluded that COP1 dimerizes via its coiled-coil domain. In our dimeric, substrate-bound datasets including the DDD-Ube2e2-COP1-c-Jun and DDD-Ube2e2-COP1-c-Jun-STK40 complexes, we now resolved part of the coiled-coil (residues H257-G301), consistent with coiled-coil mediated dimerization, while remaining segments are flexible relative to WD40 domains (revised Figure 4; Figure S10).

7. Comparison with Recent Literature

It is crucial to assess whether the proposed structural dynamics of DET1 and the assembly of DDD/UBE2Es align with recent findings by Burgess *et al.* (Science Advances, 2025).

Addressing these questions would further solidify the mechanistic insights presented in this important study.

We appreciate your suggestion and have thoroughly compared our results with those of Burgess *et al.* At the time of our original submission, their study was shared as a preprint on BioRxiv, and we discussed their findings in our Discussion section. Both studies reveal dynamic conformations of DET1 and underscore the importance of DDA1 and E2 binding in stabilizing DET1. Our study further extends these insights by providing additional cryo-EM reconstructions that include COP1, Ube2e2, and c-Jun or other substrates, enabling us to propose detailed mechanistic models for assembly and substrate-induced activation that complement and extend the observations of Burgess *et al.* We have updated the Introduction and Discussion (lines 106-111, lines 392-401) to reflect a detailed comparison with their work and cited their paper.

We are grateful for the reviewer's insightful comments, which have undoubtedly strengthened the quality and clarity of our work.

Reviewer #2 (Remarks to the Author):

The authors present novel and very interesting data on DET1 -COP1 complexes in humans. Several degrees of structural complexity are provided, association with different Ube2e proteins are described and an interesting hypothesis for substrate association and complex dissociation is proposed. In the overall the structural data is novel and opens the door to new hypothesis and possibilities in the field. However, I have some concerns given below.

Dear Reviewer #2,

Thank you for your careful review and positive feedback.

1) The work by Burgess et al, has been uploaded in bioRxiv in June 2024 and recently published in Science Advances. However, it is not being properly referenced in the text and doesn't appear in the reference list. It should also be mentioned in introduction and properly discussed. The authors could also mention on what they advance the knowledge regarding the data in Burgess et al., 2025, Science Advances. For instance, that for the first time a DET1-COP1 interaction and the DDD complex+COP1 is being seen and doesn't rely in prediction, besides all the supra-molecular complex structures described.

Thank you for your suggestion. We have now properly cited Burgess et al., 2025, Science Advances (PMID: 40009677) in the reference list and referenced their work explicitly in both the Introduction (lines 106-111) and Discussion (lines 392-401). As discussed, our study advances the field by providing, for the first time, direct structural evidence for the DET1-COP1 interaction and the assembly of the human DDD complex with COP1/substrates, extending beyond computational predictions. Furthermore, our work reveals multiple higher-order assembly states and explores substrate-induced conformational changes that were not addressed by Burgess et al.

2) Most of the molecular structures are lacking biochemical validation, and artefact formation should be discarded. Interactions with protein mutant variants should be performed to validate the structural data as the dimeric/mirror DDD and the stacking structures. Perhaps gel filtration assays with these mutated protein versions might also help.

In response, we engineered point mutations at key interaction sites within COP1-COP1 dimer, DET1-COP1 and DET1-Ube2e2 and analyzed their interactions using pull-down assays (Figure R7). However, for COP1 in the stacked assembly, the intermolecular network is quite complex, involving coiled-coil and RING-RING dimer interactions. This complexity makes it challenging to abolish octamer formation with simple mutations. To address this, we focused on specific sites identified in the COP1 dimer prediction model, which aligns well with our structural data. Several residues mediate dimer formation via hydrogen bonds and salt bridges, including R357-E266, E262-R349/K345, S397-Q260, E368/E285-K275, and E290-R276/R272 (Figure R7A). We designed mutants such as K275E/R357D and Q260A/Q262A/E290A and expressed them using combinations of GST, TSF, or HA tags. Neither mutation resulted in an obvious disruption of COP1 dimerization. Even partial deletion of the coiled-coil region (residues 234-305 or 251-305) did not abrogate dimerization, although expression levels were substantially decreased (Figure R7B-E). Due to protein instability caused by these mutations, it was not feasible to purify sufficient material for gel filtration analysis. In parallel, structure-guided mutations at the DET1-COP1 and DET1-UBE2E2 interfaces validated those interactions (Figure R7F-J).

Figure R7: Pull-down assays of COP1 coiled-coil domain and interaction of COP1-DET1 and DET-Ube2e2.

A, The AlphaFold3 model of the COP1 dimer. Close-up view of the molecular interface between COP1 coiled-coil domain are shown below. The key residues contributing to the interaction are labelled. **B-E**, Pull-down experiment of wild type or key binding residues mutants or truncation of GST-COP1 and TSF-COP1 (**B**), TSF-COP1 and HA-COP1 (**C and D**) and His-DDB1, DDA1, DET1, TSF-COP1 and HA-COP1 (**E**), the eluents were analyzed by western blotting. **F-G**, Close-up view of the molecular interface between DET1 and COP1^{WD40a} or COP1^{WD40b}. The key residues contributing to the interaction are labelled. **H**, Pull-down experiment of wild type or key binding residues mutants of TSF-DET1, GST-COP1 and His-Ube2e2, the eluents were analyzed by western blotting. **I**, Close-up view of the molecular interface between DET1 and Ube2e2, with the key interacting residues labelled. **J**, Pull-down

experiment of wild-type and key binding site mutants of TSF-DET1 and GST-Ube2e2. The eluents were analyzed by western blotting. TSF, twin-strep-FLAG tag.

3) Additionally, despite COP1-c-Jun interaction was biochemically explored, there is no additional biochemical evidence that the association with c-Jun triggers the complex dissociation.

Thank you for noting this point. We have applied our purified DDD-COP1 and DDD-Ube2e2-COP1-c-Jun complexes to a Superose 6 Increase 10/300 GL column for analysis (Figure R8A). For the DDD-COP1 protein, two distinct peaks corresponding to stacked filaments and the DDD or DDD-E2 complex were observed. In contrast, analysis of the DDD-COP1-Ube2e2-c-Jun complex revealed three peaks, including 1) aggregates and stacked filaments, 2) dimeric organization and 3) the DDD or DDD-Ube2e2 complex (Figure R8B-F). Our results indicate that co-transfection of c-Jun with the DDDC-E2 complex in cells leads to the formation of dimeric oligomers, as shown by negative-stain EM and gel filtration. Additional substrate including ETS2 co-expressed with DDD-Ube2e2-COP1 complex could also induce dimeric assembly formation (Figure R8G-K).

Figure R8: The gel filtration results of DDD-COP1 complex and DDD-COP1-Ube2e2-c-Jun complex.

A, The gel filtration profile of the purified DDD-COP1 complex (black) and DDD-COP1-Ube2e2-c-Jun complex (red). The negative stain micrograph of different peaks were presented in **B-F**. **G-J**, The negative stain micrograph and 2D class averages of the DDD-COP1 complex (**G**) and the DDD-Ube2e2-COP1-c-Jun complex (**H**), the DDD-Ube2e2-COP1-c-Jun-STK40 complex (**I**) and the DDD-Ube2e2-COP1-ETS2 (**J**). Scale bar, 100 nm. **K**, Coomassie blue-stained SDS-PAGE analysis of the purified DDD-Ube2e2-COP1-ETS2 complex.

4) Could authors experimentally demonstrate that dimeric DDD-COP1 complex is inactive whereas the monomeric DDD-COP1 complex is active?

Based on additional experiments with substrate-bound DDD-Ube2e2-COP1 complexes, we propose that the stacked assembly likely represents an inhibited state, while the dimeric DDD-Ube2e2-COP1 is active, supported by co-expression/purification and negative stain EM analysis. Consistently, co-expression with COP1 substrates, including c-Jun, c-Jun/STK40 or ETS2 resulted in a dimeric assembly of the DDD-Ube2e2-COP1-substrates complex. We further purified the strep-eluted DDD-Ube2e2-COP1-c-Jun complex using Superose 6 Increased 10/300 GL, collected the ~700 kDa fraction (eluted at ~11 ml), and confirmed its dimeric architecture by negative stain EM (Figure R9, *left and middle*). In parallel, *in vitro* ubiquitination assays using this fraction, in the presence of E1, ATP, ubiquitin, Ube2d3 and neddylated CULLIN4-RBX1, showed robust c-Jun polyubiquitination by western blot (Figure R9, *right*).

Figure R9: *In-vitro* ubiquitination reaction using the dimeric DDD-Ube2e2-COP1-cJun complex. Strep-eluted DDD-Ube2e2-COP1-c-Jun complex, or dimeric complex purified by Superose 6 size exclusion chromatography were incubated with E1 (Uba1), ubiquitin, Ube2d3 and neddylated CULLIN4-RBX1. Immunoblotting with antibodies against c-Jun and ubiquitin was performed, with the presence (●) or absence (x) of each component indicated. The experiments were repeated independently three times with similar results.

5) c-Jun doesn't appear in the images captured with the DDD complex. What is the evidence that the protein is still there and intact when the images are collected?

We routinely apply freshly purified protein complexes directly to cryo-EM grids; additionally, we have stored and thawed samples of DDD-COP1-Ube2e2-c-Jun for crosslinking mass spectrometry using the cross-linker DSBU (Figure R10A). The presence of c-Jun is confirmed by crosslinking evidence showing that intermolecular cross-links with COP1 were detected (Figure R10B), and SDS-PAGE, supporting that c-Jun remains bound and intact throughout the workflow. We therefore attribute its invisibility in our reconstructions to conformational flexibility or disorder, not to dissociation or degradation. We have clarified this explanation in the revised Results (revised Figure 4F; lines 312-318, Table 4).

Figure R10: The XL-MS results of DDD-Ube2e2-COP1-c-Jun complex.

A, Circular plot of cross-linked DDD-Ube2e2-COP1-c-Jun subunits identified using XL-MS. Purple lines represent intra-links, whereas inter-links are colored in green. **B**, The interaction site between COP1 and the c-Jun peptide segment in XL-MS.

6) It is not fully understandable how the authors can extract the conclusions described between lines 298 and line 307. First, there is little difference between lines 6 and 7 and these differences might be due to the fact that in line 7 there is a sum of the effects of lines 5 and 6. In addition, the two-step hypothesis described in line 305/306 in which DET1 collaborates with COP1 to recruit Ube2d3, would, in base of the experiment (Fig. 5), be unnecessary, since COP1 can promote c-Jun polyubiquitination without DET1.

Then, the demonstration claimed in line 319 regarding the priming of ubiquitin in substrates lacks a solid basis.

We appreciate this feedback on our interpretation of the ubiquitination assays (Figure 5). While our results show that COP1 alone is capable of mediating c-Jun polyubiquitination in the presence of Ube2d3 but the anti-ubiquitin results demonstrated that this reaction is not as robust as in the full $CRL4^{DET1-COP1}$ complex, DET1 bridges Ube2e2, and DDB1 recruits CULLIN4-RBX1 for Ube2d3-mediated elongation. To avoid overinterpretation, we have revised the Results (lines 363-369), tempered our claim that the stepwise priming model for DET1 is still a working hypothesis and requires further validation.

7) It is not fully understood why the E1 enzyme is not necessary for the first step (ubiquitin activation) in the *in vitro* ubiquitination assays shown in Figure 5.

We apologize for any confusion. The E1 ubiquitin-activating enzyme (Uba1) was included in all *in vitro* ubiquitination reaction mixtures except lane 1, which served as our negative control.

(8) The authors mention at the end of the Discussion (lines 363-364) “Future research should investigate the interplay of COP1 with a broader range of substrates to determine how

structural transitions are fine-tuned for substrate specificity.” This might include p53. Could the structural models solved by the authors help to determine whether whether p53 is a bona fide substrate of COP1 *in vivo*?

Thank you for raising this interesting point. We are interested in how COP1 recruits p53 as well and have tested whether COP1 or DDD-COP1 directly interacts with p53 using pull-down assays and co-expression in human cells, and found no detectable interaction under the conditions tested (Figure R11). In contrast, strong binding was observed with c-Jun in parallel experiments. This suggests that direct COP1–p53 recognition may require additional cofactors, adaptors, or specific cellular signaling conditions such as glucose induce assembly of CRL4^{COP1} for p53 degradation, as suggested in Su et al. (Mol Cell, 2023, PMID: 37390815). We have this result in our Discussion section (Figure S14, lines 457-463).

Figure R11. Pull-down experiment of MBP-p53 or MBP-c-Jun in cells co-transfected with GST-COP1 or GST-COP1, His-DDB1, TSF-DET1, DDA1.

Cell lysates were divided in two fractions and incubated with GST and MBP beads. The eluents were analyzed by SDS-PAGE and western blotting. TSF, twin-strep-FLAG tag.

Minor comments:

Paragraph between **lines 231 and 252** cuts a main idea in the text and could perhaps be moved to another position.

We appreciate the reviewer’s suggestion regarding the paragraph spanning lines 231–252. After incorporating the new data, we reassessed the placement and found that this paragraph now functions as a necessary bridge between the preceding structural observations and the subsequent results (Figures 2-3, Figure S6). Moving it to another section created discontinuities and required repetition of information introduced immediately before the new panels.

Line 76. CRLs should be corrected for CRL4s.

Corrected.

Line 84 – The provided references do not provide evidence for DDA1 participation in the CDDD complex in plants neither of its involvement in photomorphogenesis.

Thank you. We have corrected the references (PMID: 15342494&17452440) and now cite appropriate literature with evidence for DDA1 functions in plants and photomorphogenesis.

Line 99 – The correct reference for COP1-DET1 interaction in plants is Cañibano et al., 2021, Molecular Plant

Thank you for catching this. We now cite Cañibano et al., 2021, Molecular Plant, as the correct reference.

Line 142 – Which previous studies? References are missing. The use of references could be checked across the work.

We have added specific references (PMID: 31693911, 31686031, 38513086, 30564455) at this position and reviewed citations throughout the manuscript for completeness.

We sincerely appreciate your suggestion, which has enhanced the mechanistic depth of our study, and have incorporated all new data and analyses into the revised manuscript.

Reviewer #3 (Remarks to the Author):

Dear Reviewer #3,

We appreciate your helpful comments, which have significantly improved our manuscript.

Reviewer #4 (Remarks to the Author):

This paper presents the cryo-EM structures of the human COP1-DET1 ubiquitin ligase complex, providing insights into the distinct assembly states of COP1 in complex with the DDB1-DDA1-DET1 (DDD) complex and a potential E2 enzyme. While the study provides valuable structural data, several critical issues need to be addressed before the paper can be accepted.

Dear Reviewer #4,

Thank you for taking the time to review our work and for your positive comments.

The primary concern lies in the structural data processing and modeling. The authors provide only two maps out of eight, which raises questions about the completeness and quality of the data. Despite claims of a 2.65 Å reconstruction for the DDD complex, the DET1 component is poorly resolved, even at lower thresholds. At higher thresholds, over 50% of the potential is missing, and there are significant issues with the Ramachandran plot, which shows 12 outliers. These concerns are compounded by the fact that only partial data has been shared, and the maps presented appear suboptimal.

We are sorry for the incomplete cryo-EM data submission in our original manuscript due to an oversight. In this revised submission, we have now included the complete set of cryo-EM data used in this study, including a new data on the DDD-Ube2e2-COP1-c-Jun-STK40 complex which we collected and analyzed during revision. This included all unsharpened maps, associated masks, half-maps, B-factor sharpened and DeepEMhancer postprocessed maps used for model building and visualization. We hope these additional data will facilitate thorough evaluation and independent verification of our work. Additionally, we have reviewed and refined our structural models to reduce the number of outliers and improve overall model quality (Ramachandran, MolProbity etc.). We believe these improvements enhance the rigor and transparency, and reproducibility of our study.

The authors mention using DeepEMhancer for post-processing the maps, but it is crucial to clarify where exactly these processed maps are shown in the manuscript. Since these maps are not fully derived from the raw data, they should not be deposited to the database. Instead, the raw maps, along with the sharpened post-processed versions, should be deposited for review.

The DeepEMhancer post-processed maps were used solely for model building and for figure generation. In accordance with best practices, all the raw unsharpened map, along with the associated half-maps and masks have been deposited to the EMDB for transparency. We have revised the figure legends and Methods to state explicitly which map type is shown.

Moreover, Figure S1, which shows the 2.65 Å map without DeepEMhancer, is of poor quality. In Figure S3, the local resolution scales for the middle and right structures are inconsistent. In Figure S4, the additional potential observed seems to fit any E2, not a specific one, and in Figure 2, the map in panel B does not support the modeling of the coiled coil. Given these issues, it is difficult to fully trust the accuracy of the structural determination, which appears to lack sufficient quality control.

Regarding Figure S1, we acknowledge the challenge of resolving DET1 due to its highly dynamic nature. The DET1 density is not easily resolved when in complex with DDB1-DDA1

alone. Therefore, our structural analysis focused on the DDD-COP1 and DDD-Ube2e2-COP1-c-Jun complexes, which enabled more reliable modelling of DET1.

For Figure S3, we have revised the figure so that both panels now display local resolution scales with consistent color gradients.

In Figure S4, we detected endogenous E2 proteins that co-purified with the DDD-COP1 complex. Mass spectrometry confirmed the presence of all Ube2e family members and Ube2d3, each sharing a similar UBC domains. Previous literature supports DET1 binding to the Ube2e family, and accordingly, we modeled Ube2e2 in our structure with trimmed side chains to reflect this ambiguity. We also included Ube2e2 in subsequent co-expression experiment to further support this assignment.

Regarding Figure 2 (now Figure 1), we have performed local refinement and provided DeepEMhancer-postprocessed maps of both layer 1 and layer 2 and presented these maps in Figure S3. These provide additional support for our modelling of the coiled-coil region.

Taken together, we hope these revisions and additional experiments address the reviewer's concerns and strengthen the accuracy and transparency of our structural determination. We thank the reviewer for this valuable feedback.

Additional comments:

- Introduction: **Line 77** refers to CRLs consisting of a scaffold subunit CULLIN4, but it is important to note that there are at least nine Cullin proteins in the CRL family. A summary paragraph at the end of the introduction, such as "In this study, we used single-particle cryo-EM to...", would help clarify the study's aims and approach.

Thank you for this suggestion. The introduction now acknowledges that CRLs comprise at least nine different cullin scaffold proteins (revised lines 77). We have also added a concluding paragraph (revised lines 131-139) to the Introduction summarizing our aim: "In this study, we used single particle cryo-EM to elucidate the structure of human COP1 in complex with DDB1-DDA1-DET1 (the DDD module) and Ube2e2, revealing stacked like assembly state. Our findings show that DET1 serves as a structural scaffold, connecting COP1 to DDD and Ube2e2 enzymes, while DDB1 recruits the CULLIN4-RBX1 complex. Substrates of COP1 including c-Jun or ETS2 could induce conformational change within the assembly. This, in turn, enables engagement of additional E2 enzymes via RBX1 to promote polyubiquitination. The study highlights the dynamic structural interplay within the CRL4^{DET1-COP1} E3 ligase complex and its activation upon substrate binding, offering key insights into substrate ubiquitination and the higher-order assembly mechanisms of E3 ligases.

- Figures and Structural Models: Figure 1B should be cited earlier when discussing the domains of DDB1. The authors present many structural models but fail to provide sufficient cryo-EM maps to support these models, which weakens the validity of the claims.

We have now referenced Figure 1B at the appropriate location where the domain structure of DDB1 is first described (revised line 148, 155) and the complete set of cryo-EM data resolved in this study have been provided with this submission.

- **Line 191:** The statement about C3 symmetry is inaccurate. A revision is needed.

We apologize for the oversight. We have revised this part.

- **Line 202:** This paragraph is unclear. It is unnecessary to superimpose models to predict zinc fingers, as AlphaFold typically provides accurate predictions. If AlphaFold models were used, this should be stated clearly.

Thank you for pointing this out. In our study, we have done structure alignment between the TRIM21 and COP1 RING domains that built from the stacked assembly to illustrate the conserved zinc finger motifs. Upon review, we feel this panel added limited value and could be confusing. We have therefore removed it and revised the text accordingly.

- **Line 223:** The content of this paragraph should be moved to the discussion, as it feels out of place in the results section.

Thank you for this suggestion. We agree and have moved this content from the Results to the Discussion section for better logical flow (now lines 417-423).

- **Figure 3D:** The map quality does not appear to support side chain assignments, and this should be addressed.

In Figure 3D and the corresponding text, while most secondary structure can be confidently assigned, specific side-chain placement is tentative in lower-resolution regions. To assess intermolecular contacts without overinterpretation, we aligned the predicted model to the experiment model and based interaction analysis on that alignment. We have updated in the Methods to reflect these limitations. And for the modelling, the poly-alanine model was used where justified.

- **Line 247:** The authors justify mutating to glycines instead of alanine. Glycine mutations can drastically alter protein structure, so this decision should be more thoroughly explained.

We appreciate this point. Glycine substitutions were initially chosen to disrupt side-chain mediated contacts without introducing bulky residues, which could potentially cause more pronounced local destabilization than alanine. To address this concern, we have included additional mutagenesis experiments using alanine substitutions at key positions, obtaining similar biochemical results (see revised Figure 3E and Methods), except that mutation of Ube2e2 R88A/R124A/R126A resulted in protein instability. This is now described in the Results.

Figure R12. Pull down assays of Ube2e2 and DET1.

Cell lysates were divided in two fractions and incubated with GST and MBP beads. The eluents were analyzed by SDS-PAGE and western blotting. TSF, twin-strep-FLAG tag.

• **Figure 5:** An anti-ubiquitin blot should be included to strengthen the findings, as it is critical for validating the ubiquitination results.

As recommended, we have now included anti-ubiquitin blots as part of revised Figure 5 and S7, clearly demonstrating substrate ubiquitination. The updated figure and figure legend are cited and discussed in the revised Results (lines370-375).

Figure R13: *In vitro* ubiquitination assay of DDDC-Ube2e2-c-Jun or COP1-c-Jun complexes.

A, DDDC-Ube2e2-c-Jun or COP1-c-Jun complexes, purified from Expi293F cells, were incubated with E1 (Uba1), E2, ubiquitin, and neddylated CULLIN4-RBX1. Immunoblot analysis was performed using antibodies against c-Jun and ubiquitin. **B**, *In vitro* ubiquitin assay of the COP1-c-Jun complex was performed with E1 (Uba1), ubiquitin, ATP and either Ube2d3, wild type or catalytic inactive Ube2e2 mutants shown on lanes 1-3. COP1-DET1-Ube2e2-c-Jun complex was incubated with E1 (Uba1), ubiquitin and either CULLIN4-RBX1 or neddylated CULLIN4-RBX1 shown on lanes 4-7. Immunoblotting was performed using antibodies against c-Jun and ubiquitin. The presence (●) or absence (x) of each component in the reaction is indicated. The experiments were repeated independently three times with similar results.

While the study offers important structural insights into the COP1-DET1 complex, the concerns regarding the quality and completeness of the cryo-EM data and the corresponding structural modeling are significant. The authors need to provide a more rigorous analysis, including the full set of maps and improved structural refinements, to support their conclusions. I recommend addressing these issues and resubmitting the paper once the data processing and modeling have been adequately corrected.

In summary, we thank the reviewer for these detailed and constructive comments. We have now provided the complete map set, corrected all figure and text issues, and carefully improved model validation and documentation to address all concerns. We hope these improvements fully satisfy your requirements for acceptance.

We thank the reviewers for their time and constructive suggestions to further improve the manuscript. Below is a point-by-point response to all the comments. The corresponding changes in the revised manuscript are highlighted in blue.

REVIEWER COMMENTS

Reviewer #1 (Remarks to the Author):

My concerns have been addressed.

Dear Reviewer #1,

Thank you for your thoughtful review and positive evaluation on our work. Much appreciated!

Reviewer #4 (Remarks to the Author):

The authors have made a genuine effort to improve the clarity and presentation of the manuscript in response to my previous comments. The revised version is notably clearer in both the writing and the organization. However, I still have serious concerns regarding the quality and reliability of the structural data, which directly impact the central claims of the manuscript. Unfortunately, the cryo-EM map quality remains insufficient to support the modeling and interpretations made in the paper.

Dear Reviewer #4,

Thank you for your time and careful assessment of our structural data. In the following points, we aim to provide detailed clarifications on our data processing, validation, and modelling approaches. We have also undertaken further model refinement based on your feedback, which has improved the validation metrics.

Below are specific concerns for each reported structure:

1. 9LTJ: A significant portion of DET1 is poorly resolved. The authors used an extremely low threshold value in the EM validation server, making the map appear to envelop nearly all atoms — a misleading practice that inflates interpretability.

- DET1 flexibility: We acknowledge the reviewer's point that portions of DET1 are not resolved and would like to clarify that this is a known feature of the DET1 subunit, which is flexible, particularly in the regions that are not resolved in our reconstruction. This observation is consistent with previously published structures (PMIDs: 40009677; 39937864). To ensure our model does not overstate the data, we have taken a conservative approach: we restricted side-chain modelling to well-resolved regions and pruned poorly supported side-chains to C_β while retaining amino acid assignments. We have clarified this limitation in the Materials and Methods section (lines 706-708).

- EM map thresholding:
We thank the reviewer for this insightful comment.

We have re-visualized DET1 at a higher contour level and now include panels at both lower and higher contour levels in the revised manuscript, enabling readers to assess the map quality, particularly in ambiguous regions. We also indicate the limitations in map resolution for specific parts of DET1. The recommended contour level (0.16) in 9LTJ was chosen in a way to exclude much of the noise and at the same time cover all parts of the modelled structure (*shown below*).

The unsharpened map of 9LTJ shown in different contour levels.

As noted by the reviewer, a significant portion of DET1 is indeed poorly resolved in 9LTJ (due to its flexibility); however, we have obtained improved density for this region in complexes with the E2 protein (structures 9LTO and 9LTZ), where these dynamic regions of DET1 get stabilized and are better resolved. The modelling of the poorly resolved regions in DET1 is therefore informed by multiple data sources, integrating information from these additional structures. This supporting data is presented in Supplementary Figure S3 (*shown below*) and detailed in the text in lines 157-158, 191-192.

Comparison of the unsharpened map of DDD (9LTJ) and DDD-E2 complexes (9LTO and 9LTZ).

The reported resolution is inconsistent (3.25 Å in the validation report vs. 2.65 Å in the manuscript). Moreover, the model built on the 2.65 Å map contains 3.28% rotamer outliers and Ramachandran violations, which is unacceptable for such a resolution claim.

- Inconsistent resolution:** We would like to clarify the apparent discrepancy in the reported resolution values, which stems from the standard use of both unmasked and masked FSC calculations. The 3.25 Å (probably the reviewer is referring to 3.4 Å) resolution noted in the EMDB validation report is calculated by the server on the unmasked map. This value is entirely consistent with the unmasked resolution calculated in CryoSPARC, which we provided in Figure S1E (*shown below*). The EMDB deposition server does not have a function to apply a user-provided mask for its internal FSC calculation. The final, masked resolution reported in our manuscript reflects the accepted standard in the field.

GS-FSC for DET1-DDB1-DDA1 complex (9LTJ, shown in supplementary S1E).

- Rotamer outliers: Thank you for pointing this out. We have gone through the model and performed additional real-space refinement, which has improved the geometry. The percentage of rotamer outliers and Ramachandran violations have been reduced from 3.28% to 0.12% and 0.42 to 0.08 (1 residue), respectively. The updated model has been deposited.

2. 9LTW: The sharpened map is visually almost identical to 9LTJ. The unsharpened map shows some additional density, but it is far too ambiguous to support confident placement of anything.

We thank the reviewer for carefully inspecting these maps. 9LTW and 9LTJ represent the DDD (DET1-DDB1-DDA1) subcomplex as classified from within larger, more heterogeneous particle populations. Specifically, 9LTW represents the DDD complex found within the DDB1-DDA1-DET1-COP1-Ube2e2-c-Jun dataset, which is why its map appears nearly identical to 9LTJ. While these maps showed limited interpretability due to this inherent heterogeneity and preferred orientation, we decided to deposit them as individual entries and built the model only in regions supported by density. This ensures full transparency of our data processing pipeline and accurately reflects the compositional variability discovered within our samples.

3. 9LTZ: The map appears streaky and lacks interpretable features. Manual modeling would not be feasible without AlphaFold models, calling into question the legitimacy of the model building.

9LTZ represents the DDD-E2 subpopulation as classified from the DDB1-DDA1-DET1-COP1-Ube2e2-c-Jun dataset, which corresponds to the same particle class as 9LTO in (DDB1-DDA1-DET1-COP1) collection. The 9LTZ is of limited interpretability due to preferred orientation and particle heterogeneity; we have deposited it as part of our commitment to full data transparency, but we want to be clear that it does not form the basis for our conclusions. For all presentation and interpretation of the DDD-E2 assembly, we therefore rely on the higher-quality reconstruction (9LTO) and model only in regions supported by clear density.

While some preferred orientation was observed in the datasets, we could build the models by using AlphaFold2 predictions and previously reported structures as reliable guides. AlphaFold2 predicted models are widely used as a high-quality foundation for structural analysis. By enabling the fitting of AlphaFold2 models onto cryo-EM maps, AlphaFold2 facilitates precise protein positioning, significantly enhancing the quality of the final models. This is particularly beneficial when working with low-resolution data, by providing accurate initial models to fit structural components into density maps (reviewed in PMIDs: 39313455, 39133843).

In this work, AlphaFold2 predictions were used as initial models when actual experimental models are not available. Critically, the final coordinates of our models were refined and constrained strictly by the experimentally observed density. This ensures that the final model

is a faithful representation of our data, not the prediction. This hybrid approach is now generally used for dynamic or challenging targets (e.g. Xia et al, *Nature Communications*, PMID: 41136424; Kulczyk et al, *Nature Communications*, PMID: 40769979; Jang et al, *Nature Communications*).

Furthermore, to directly and quantitatively address the reviewer's concern about data quality, we have performed a new anisotropy analysis for all structures in this study using the Orientation Diagnostics tool in CryoSPARC. While the global resolutions of the structures achieved are between 2.65 – 4.33 Å, we acknowledge that, because of orientation bias, the resolution in the least-resolved direction is lower than the global average. The corresponding cFAR scores which quantify this effect are provided in the Supplementary Table S1 and stated in lines 694-698. To complement cFAR, we also examined the sampling compensation factor (SCF) for each reconstruction. These data are now presented in Supplementary Table S1, providing a full and transparent account of the directional resolution of our maps.

4. 9LU1: This is the only map in the set that is of generally acceptable quality. However, it still barely supports the modeling in Fig. 4.

We are pleased that the reviewer finds the quality of this map acceptable. To aid in visualizing the key features shown in Fig. 4, we used DeepEMhancer post-processing map for model building. However, we want to clarify that all model refinement and validation (including map-model FSC and CCmask) were performed exclusively against the original, un-filtered cryo-EM map. Where density was ambiguous, we have removed amino acid side chains to avoid over-interpretation.

5. DDD-ube2e2-COP1-c-Jun-STK40: It is unclear which PDB entry this corresponds to. The validation report provided shows "NOT for manuscript review", and again, the thresholding appears artificially low. The map is extremely streaky and the backbone is not traceable (also impossible for the deep-enhanced model). This pattern is suggestive of severe preferred orientation.

We thank the reviewer for your comments on this dataset. First, we would like to clarify its identity and context. This dataset corresponds to a new complex generated specifically to address a question raised by Reviewer #1 in the previous round of revision, who asked whether the c-Jun adaptor STK40 might stabilize c-Jun for cryo-EM visualization. We are pleased to report that this additional experiment successfully addressed the original query.

We are sorry for the preliminary "NOT for manuscript review" validation report submitted previously. The official wwPDB validation report has been provided.

Accordingly, our modeling approach has been conservative: the protein models were rigid body fitted, the coordinates were placed only where supported by experimental density, and we make no claims about side-chain interactions from this map. We have provided this clarification in Materials and Methods section of the paper (lines 706-708). In addition, the Orientation Diagnostics for this reconstruction have been provided in Supplementary Table S1.

Crucially, the central conclusions of our manuscript do not depend on this dataset. We have included it to transparently document the experimental attempt to answer a reviewer-raised biological question. While the map quality is not ideal, we believe presenting this data is the most rigorous approach, as it remains highly relevant to the work.

6. 9LTO: Although the additional density is less streaky than others, the placement of an E2 is still speculative at best, with no clear features supporting its orientation or conformation.

We acknowledge that 9LTO may not permit de-novo tracing of Ube2e2. However, the density clearly resolves the secondary structure elements, which allowed for an unambiguous rigid-body placement of the model. We confirmed this placement by carefully inspected the cryo-EM density and considered alternative model orientations, with the presented model representing the best fit (*shown below*). This was explained in the manuscript (lines 183-186).

Fitting of E2 (Ube2e) in the cryo-EM density shown in two orientations.

7. 9LTL: This map does not allow tracing of the main chain, yet the authors report a resolution of 2.93 Å — a value that is clearly inconsistent with the visible density.

For these reconstructions, we acknowledge the reviewer's observation that there can be a discrepancy between the global FSC resolution and the visual interpretability of certain regions. This is a common challenge for large, dynamic, or multi-component complexes that exhibit significant compositional and conformational heterogeneity.

9LTL represents the DDD complex within the DDB1-DDA1-DET1-COP1 collection. On the reported “2.93 Å” versus visible density. The 2.93 Å value is the global, masked gold-standard FSC (0.143) resolution for the map. However, this dataset exhibits pronounced heterogeneity and anisotropy (driven by preferred orientation and compositional variability), yielding a broad local-resolution range. While the core shows local resolution approaching ~3.0–3.5 Å, substantial portions of the map are >5–8 Å and do not support main-chain tracing. This explains the apparent inconsistency between the global number and the visual interpretability across the map. We have provided this clarification in M&M section of the paper (lines 693-694).

8. 9LTR: The map is again streaky, likely due to preferred orientation and inclusion of low-quality particles. The reported resolution of 3.03 Å appears greatly exaggerated.

Following standard practice, the resolution was determined by the gold-standard Fourier Shell Correlation (FSC) 0.143 cut-off. This was calculated between two independently refined half-maps as implemented in CryoSPARC. However, we acknowledged the preferred orientation and in lines 693-698 of the revised manuscript and provided detailed orientation analyses in the Supplementary Table S1.

9. 9LUL, 9M0Y: While the coiled-coils are faintly visible, there is no meaningful detail beyond the global shape. The reported resolutions in the 4.x Å range do not reflect the poor interpretability of these maps.

Coiled coil region of COP1 is dynamic, therefore this region is visible at low contour levels, we have therefore restricted our modelling to only backbone fitting, supported by Alphafold2 predictions. While density is faint, it is observable, and supports the placement of coiled-coil within the complex. The experimental model agrees well with the predicted Alphafold2 model, providing additional level of confidence. This was as well commented in the main text (lines 710-711).

10. Supplemental Fig. S9. The FSC plots lack numerical resolution values, which limits their interpretability. In panels A, B, and D–G, more than half of the frequency components are zero, suggesting that unnecessary large boxes were used for the analysis. Additionally, in panel C, the FSC curve does not drop to zero at high frequency, which raises concerns about potential particle duplication or overfitting.

Supplementary Fig. S9 was updated as Fig. S12 during the last round of revision. Probably the reviewer is referencing the previous version of the manuscript. We have gone through and revised the models, as well as repeating real-space refinement jobs. All the model-to-map FSC plots were updated in the last revision. We have regenerated the plot, which now correctly drops to zero at high frequency, and have updated the figure in the revised manuscript.

11. General Recommendation: I strongly recommend that the authors seek guidance from an experienced cryo-EM structural biologist regarding data processing, map validation, and model building. The quality of both maps and models is foundational to structural biology, and in its current form, the manuscript does not meet the necessary standards for publication.

Despite improved clarity in writing, the structural data remain insufficient in quality and reliability. Therefore, I cannot support publication of this work in its current form.

We have followed community best practices for cryo-EM data analysis and validation, and are confident in our data and conclusions. We believe the manuscript has been significantly strengthened by this review process and thank the reviewer for the time and effort.

Beyond the specific technical points addressed above, we wish to emphasize the integrated structural and functional approach that underpins our study's conclusions. We acknowledge that reconstructing dynamic complexes can yield maps that are informative but not always perfectly resolved in every region. For this reason, our study was designed to test the functional hypotheses derived from our structural models. We embarked on biochemical validation through mutagenesis. While such experiments are challenging, their success here provides additional evidence for the functional relevance of the interfaces we modeled. Importantly, while we have improved the technical robustness of our map and model derivations in this revised submission in response to the reviewer's feedback, our central biological conclusions remain entirely unchanged. This hope this underscores the reliability of our initial interpretation and demonstrates that the functional significance we derived from our structural biology results is sound.

We again would like to thank all reviewers for their time.

Reviewer #4 (Remarks to the Author):

Thank you for providing additional explanations and clarifications in response to my previous comments. At this stage, I do not have further constructive suggestions to offer. However, I remain concerned about the overall quality and interpretability of several cryo-EM maps and the degree to which the structural models rely on AlphaFold predictions. While I appreciate the effort made in this revision, these issues continue to limit my confidence in the structural conclusions.

Dear Reviewer #4,

Thank you again for your time and previous helpful suggestions.